# Mst1-mediated phosphorylation of FoxO1 and C/EBP-β stimulates cell-protective mechanisms in cardiomyocytes

Yasuhiro Maejima[1,2], Jihoon Nah[1,3], Zahra Aryan[4], Peiyong Zhai[1], Eun-Ah Sung [1], Tong Liu[5], Koichiro Takayama [1], Siavash Moghadami[6], Tetsuo Sasano[2], Hong Li[5] & Junichi Sadoshima [1] ✉

The molecular mechanisms by which FoxO transcription factors mediate diametrically opposite cellular responses, namely death and survival, remain unknown. Here we show that Mst1 phosphorylates FoxO1 Ser209/Ser215/Ser218/Thr228/Ser232/Ser243, thereby inhibiting FoxO1-mediated transcription of proapoptotic genes. On the other hand, Mst1 increases FoxO1-C/EBP-β interaction and activates C/EBP-β by phosphorylating it at Thr299, thereby promoting transcription of prosurvival genes. Myocardial ischemia/reperfusion injury is larger in cardiac-specific FoxO1 knockout mice than in control mice. However, the concurrent presence of a C/EBP-β T299E phospho-mimetic mutation reduces infarct size in cardiac-specific FoxO1 knockout mice. The C/EBP-β phospho-mimetic mutant exhibits greater binding to the promoter of prosurvival genes than wild type C/EBP-β. In conclusion, phosphorylation of FoxO1 by Mst1 inhibits binding of FoxO1 to pro-apoptotic gene promoters but enhances its binding to C/EBP-β, phosphorylation of C/EBP-β, and transcription of prosurvival genes, which stimulate protective mechanisms in the heart.

Mammalian sterile 20-like kinase 1 (Mst1), a ubiquitously expressed serine/threonine kinase[1], is a key component of the "*Hippo*" signaling pathway, an evolutionarily conserved signaling pathway regulating organ size, tumorigenesis, and tissue regeneration[2,3]. Mst1 is activated by stresses, such as oxidative stress, and proinflammatory cytokines, through either auto-phosphorylation or caspase-mediated cleavage[4]. Upstream components of the Hippo pathway, including NF2, and another scaffolding protein, Rassf1A, are involved in the phosphorylation and activation of Mst1[4]. Both Mst1 and Mst2, a homolog of Mst1, regulate the growth and death of many cell types through either canonical or non-canonical Hippo pathways. In the canonical Hippo pathway, Mst1 and Mst2 phosphorylate Lats1 and Lats2, serine/threonine kinases, through Salvador- and Mob-dependent mechanisms,

which in turn inactivate Yap and Taz, transcription factor co-factors in the nucleus, through either nuclear exit or proteolysis[3]. Yap and Taz control cell proliferation and survival through interaction with transcription factors, including TEAD and Forkhead box O (FoxO) family transcription factors, which in turn mediate cell proliferation and survival[3]. Importantly, Mst1 also phosphorylates its substrates at various subcellular localizations and controls cellular functions independently of Yap or Taz, through alternative Hippo pathways collectively termed non-canonical. For example, Mst1 is activated by oxidative stress in mitochondria through a K-Ras-Rassf1A-dependent mechanism and phosphorylates Bcl-xL at Ser14 to induce dissociation of Bcl-xL from Bax and the mitochondria-dependent mechanism of apoptosis[5]. Mst1 phosphorylates Beclin1 at Thr108 in the endoplasmic reticulum to

[1]Department of Cell Biology and Molecular Medicine, Rutgers New Jersey Medical School, Newark, NJ, USA. [2]Department of Cardiovascular Medicine, Tokyo Medical and Dental University, Tokyo, Japan. [3]Department of Biochemistry, Chungbuk National University, Cheongju, Korea. [4]Department of Medicine, Rutgers New Jersey Medical School, Newark, NJ, USA. [5]Center for Advanced Proteomics Research and Department of Biochemistry and Molecular Biology, Rutgers New Jersey Medical School, Newark, NJ, USA. [6]Department of Chemical and Systems Biology, Stanford University School of Medicine, Stanford, CA, USA. ✉e-mail: sadoshju@njms.rutgers.edu

stimulate physical interaction between Beclin1 and the Bcl-2 family proteins, thereby inhibiting autophagy[6].

FoxO transcription factors are involved in a wide variety of functions in cells. The activity of FoxOs is tightly controlled by post-translational modifications, including phosphorylation, acetylation, and ubiquitination[7]. Phosphoinositide-3-kinase (PI3K)/Akt signaling-mediated phosphorylation of FoxO results in the translocation of FoxOs from the nucleus to the cytosol, rendering them inactive[8]. In addition to Akt, extracellular signal-regulated protein kinase, cyclin-dependent kinase 1 (CDK1), and CDK2 can phosphorylate and inhibit FoxO activity through nuclear exclusion and/or polyubiquitination and degradation of FoxO via the proteasome system[9–11]. On the other hand, several other kinases can promote FoxO activation by facilitating the nuclear translocation of FoxO. AMP-activated protein kinase phosphorylates FoxO3, thereby increasing target gene transcription[12]. JNK phosphorylates FoxO4 to promote nuclear translocation and activation of FoxO4 in response to reactive oxygen species, thereby inducing expression of the FoxO target genes, including MnSOD and catalase[13]. One of the most puzzling aspects of FoxO cellular function is that FoxOs regulate both cell death-promoting and -inhibiting, namely diametrically opposite, mechanisms, even in the same cells, and both salutary and detrimental functions of FoxOs have been demonstrated in vivo. We hypothesized that post-translational modification of FoxOs alters the pairing mechanisms between FoxOs and their target genes, thereby regulating different sets of genes in a context-dependent manner.

Mst1 phosphorylates both FoxO1 and FoxO3, and promotes FoxO nuclear translocation, thereby inducing cell death in neuronal cells[14,15]. On the other hand, Mst1 also promotes lifespan extension in *Caenorhabditis elegans* (*C. elegans*) through phosphorylation of *DAF-16*, an ortholog of mammalian FoxO[14]. Phenotypic analysis of systemic Mst1 knockout mice (*Mst1*-KO) revealed that the Mst1–FoxO1 signaling pathway plays a crucial role in the survival of naïve T cells[16]. Furthermore, concurrent clinical studies demonstrated that a novel primary immunodeficiency phenotype caused by progressive loss of naïve T cells is associated with a genetic deficiency of Mst1, possibly through the impairment of Mst1–FoxO signaling[17,18]. Thus, Mst1 interacts with the FoxO family transcription factors and regulates cell growth, survival, and death. We reasoned that investigating the molecular mechanisms by which Mst1 controls the diverse functions of FoxOs might provide us with a hint to understand how FoxOs control the expression of genes with diametrically opposite functions.

CCAAT/enhancer binding protein-β (C/EBP-β), a transcription factor which possesses a C-terminal basic leucine zipper (bZIP) domain and was initially identified as a gene involved in the acute response of the liver, mediates various cellular processes, including inflammation, proliferation, differentiation, and production of antioxidants, such as catalase[19,20]. C/EBP-β is also known as a negative regulator of physiological cardiac hypertrophy through repression of CM growth and proliferation[21]. C/EBP-β forms either a homodimer or heterodimer with other C/EBP family proteins or interacts with additional transcription factors, including FoxOs, CREB, NF-κB, and AP1[22,23]. The DNA-binding activity of C/EBP-β increases in response to hypoxia or ischemia/reperfusion (I/R) in various organs, including the heart, liver, and lung[24]. However, the role of C/EBP-β transcription during hypoxia or I/R and how C/EBP-β regulates gene expression in the heart remains unknown.

We here demonstrate that C/EBP-β is activated through phosphorylation in the leucine zipper domain by Mst1 through a FoxO1-dependent mechanism, thereby protecting the heart against ischemia. Although FoxO1 promotes transcription of pro-apoptotic genes, Mst1-induced phosphorylation of FoxO1 induces interaction between FoxO1 and C/EBP-β, phosphorylation of C/EBP-β, dimerization and activation of C/EBP-β, and upregulation of pro-survival genes. Our results show a molecular mechanism by which Mst1 converts FoxO1 to a cell-protective molecule through activation of C/EBP-β.

## Results

### Mst1 phosphorylates FoxO1 and enhances nuclear localization

Given the interaction between Mst1 and FoxO1 in other cell types[14], we examined whether Mst1 affects the intracellular localization of FoxO1 in CMs using immunostaining. Endogenous FoxO1 was localized diffusely in both the cytoplasm and the nucleus. Treatment with $H_2O_2$, which is known to stimulate Mst1 kinase activity[25], promoted nuclear localization of FoxO1. Similarly, transduction of an adenovirus harboring Mst1 (Ad-Mst1) induced nuclear localization of FoxO1 in CMs (Fig. 1a). These results suggest that Mst1 enhances nuclear translocation of FoxO1 in CMs.

We next examined whether endogenous Mst1 interacts with endogenous FoxO1 in CMs. Heart homogenates were subjected to immunoprecipitation with anti-Mst1 or anti-mouse IgG antibody, followed by immunoblotting with anti-FoxO1 antibody. FoxO1 was found in Mst1 immunoprecipitates at baseline and in the presence of pressure overload induced by transverse aortic constriction (TAC) for 2 weeks, a stress known to induce Mst1 activation in vivo[26]. The band detected with anti-FoxO1 antibody was not detected in samples prepared from cardiac-specific FoxO1 knockout (*FoxO1*-cKO) mice, confirming the specificity of the assay. These results suggest that Mst1 and FoxO1 physically interact with one another in the heart (Fig. 1b). The interaction of FoxO1 with Mst1 was further confirmed by in vitro pull-down assays. Recombinant full-length FoxO1 (FoxO1-FL: amino acid 1–652) and the DNA-binding domain (DBD) of FoxO1 (FoxO1-DBD: amino acid 111–280), but not other truncated mutants of FoxO1 (FoxO1-NT: amino acid 1–118, FoxO1-CT1: amino acid 280–421, FoxO1-CT2: amino acid 421–652), were able to interact with 3xFlag-Mst1-WT (Fig. 1c). Conversely, truncated mutants of Mst1 containing the kinase domain of Mst1 were able to interact with FoxO1-DBD (Supplementary Fig. 1a). These results suggest that the kinase domain of Mst1 binds directly to FoxO1-DBD.

We next examined whether Mst1 phosphorylates FoxO1. We conducted in vitro kinase assays using FoxO1-FL, FoxO1-NT, FoxO1-DBD, FoxO1-CT1, and FoxO1-CT2 as substrates (Fig. 1d). After incubation with $^{32}$P-labeled ATP, strong phosphorylation of FoxO1-FL and FoxO1-DBD and very weak phosphorylation of FoxO1-CT1 were observed in an Mst1-dependent manner, suggesting that Mst1 directly phosphorylates FoxO1, primarily in the DBD. FoxO3 and FoxO4, which possess DNA-binding domains with amino acid sequences nearly identical to that of FoxO1, also underwent facilitated nuclear translocation in a manner dependent on the kinase activity of Mst1 (Supplementary Fig. 1b, c). Previous studies have shown that Mst1-induced phosphorylation of human FoxO1/3 at Ser212/207 disrupts the interaction between FoxO1/3 and 14-3-3 proteins, an effect opposite from that of myrAkt (a myristoylated form of Akt associated with the plasma membrane that is constitutively active) upon FoxOs[14,15]. We evaluated the effect of Mst1-induced phosphorylation of FoxO1 on its association with 14-3-3 proteins. We transduced CMs with Ad-Mst1 or Ad-myrAkt together with Ad-FoxO1 and immunoprecipitated FoxO1. The FoxO1 immunoprecipitates were subjected to immunoblot with anti-14-3-3 antibodies (Supplementary Fig. 1d). Although overexpression of myr-Akt increased the ability of FoxO1 to interact with 14-3-3 proteins, FoxO1 interaction with 14-3-3 protein was decreased in Mst1-overexpressing CMs. It has been shown that Mst1 directly inhibits Akt in cancer cells[27]. Mst1 negatively regulated phosphorylation of Akt at Ser483, whereas DN-Mst1 increased Akt phosphorylation in CMs (Supplementary Fig. 1e). These results suggest that inhibition of Akt and consequent dissociation of FoxO1 from from 14-3-3 may contribute to the nuclear translocation of FoxO1 by Mst1.

Mass spectrometric analyses of the product of the in vitro kinase assay demonstrated that Ser209, Ser215, Ser218, Thr228, Ser232, and Ser243 in mouse FoxO1, located in the α3 helix to wing2 portion of the DBD, are phosphorylated by Mst1 (Fig. 1e; Supplementary Fig. 2a–c; Supplementary Note 1). These amino acids are highly conserved across

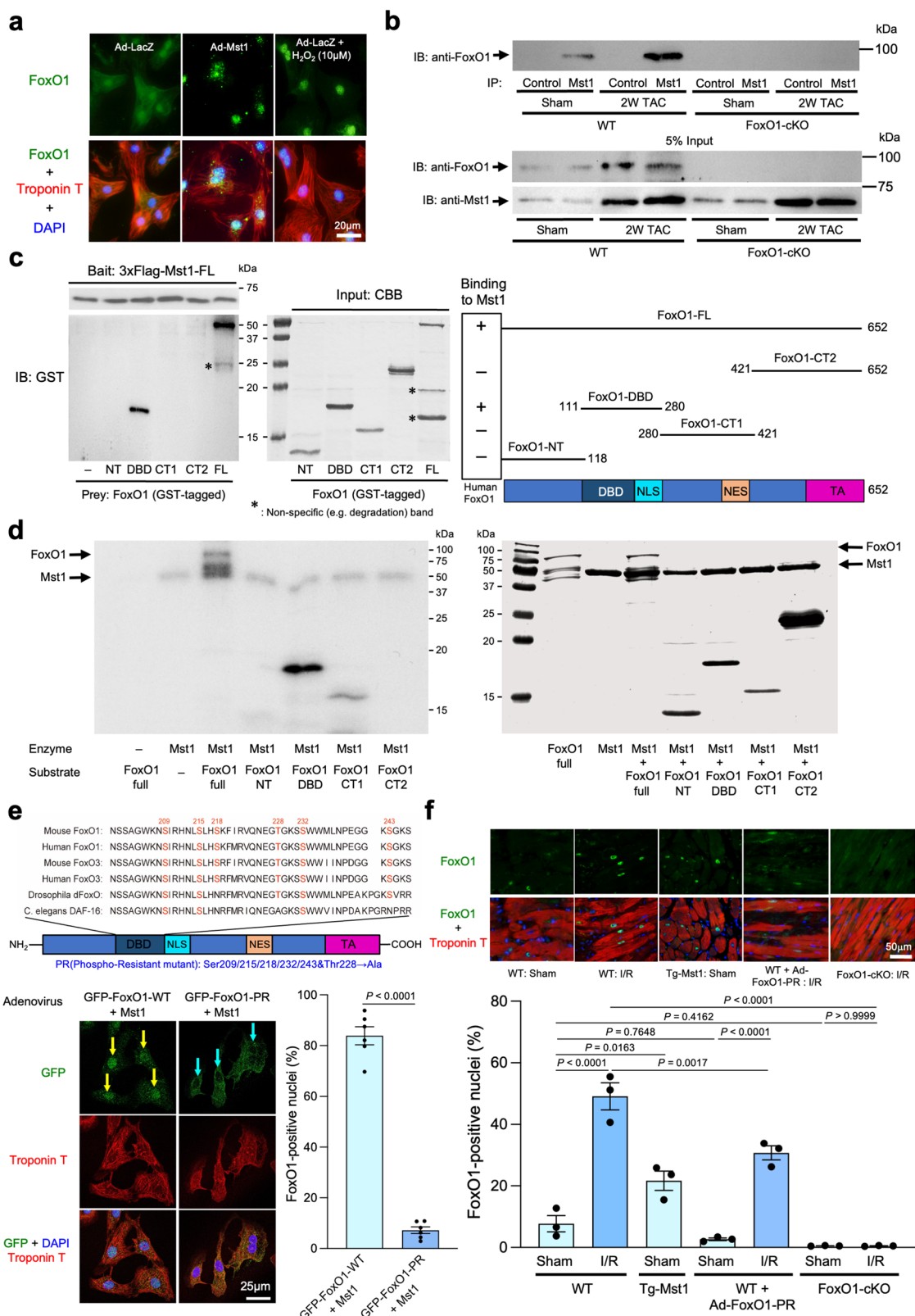

species and within the forkhead domain of the FoxO family of transcription factors, involved in DNA binding to the FoxO binding element[14,15,28]. Previous reports suggested that Mst1 phosphorylates Ser209, Ser215, Ser231, and Ser232 within the forkhead domain[14,15,28]. However, using high-resolution mass spectroscopy, we found that a slightly different set of amino acids is phosphorylated by Mst1. In order to evaluate the role of Mst1-induced FoxO1 phosphorylation as a whole, we also made an adenovirus harboring a FoxO1 phospho-resistant (PR) mutant in which Ser209, Ser215, Ser218, Thr228, Ser232, and Ser243 are replaced with alanine (FoxO1-PR). Since a previous study showed that phosphorylation of FoxO1 at Ser209 is sufficient for nuclear translocation of FoxO1[15], we made a FoxO1 mutant in which only Ser209 is mutated to Ala (S209A). We conducted immunoblot analyses using a commercially available phospho-Ser209-FoxO1 antibody. Mst1

**Fig. 1 | Mst1 phosphorylates FoxO1, thereby enhancing nuclear translocation of FoxO1. a** Cardiomyocytes treated with Ad-Mst1 or Ad-LacZ in the presence or absence of $H_2O_2$ (10 μM) were stained with a FoxO1 antibody (green), a troponin T antibody (red), and 4′,6-diamidino-2-phenylindole (DAPI; blue). **b** Coimmunoprecipitation assays with myocyte lysates. After immunoprecipitation with control IgG or an Mst1 antibody, immunoblots for endogenous FoxO1 were performed. **c** *Left:* interaction between Mst1 and FoxO1 was examined by pull-down assays with indicated recombinant proteins. *Middle:* Coomassie brilliant blue (CBB) staining of the gel after SDS–PAGE. *Right:* diagrams of mouse FoxO1 recombinant proteins (full-length and partial) used as preys of pull-down assays or substrates of in vitro kinase assays. DBD DNA-binding domain, NLS nuclear localization signal, NES nuclear export signal, TA transactivation domain. **d** *Left:* in vitro kinase assays were carried out by incubating recombinant Mst1 with recombinant mouse FoxO1 proteins (full-length and partial) as substrates. *Right:* CBB staining of the gel after SDS–PAGE. **e** *Upper:* alignment of sequences in the forkhead (DNA-binding) domain of mouse/human FoxO1 and FoxO3, *Drosophila* dFOXO, and *C. elegans* DAF-16. *Lower:* cardiomyocytes treated with adenoviruses harboring GFP-tagged wild-type FoxO1 or FoxO1-Ser209/Ser215/Ser218/Thr228/Ser232/Ser243A (PR) mutant in the presence or absence of Mst1 were stained with a GFP-tag antibody (green), a troponin T antibody (red), and DAPI (blue). *Left:* representative immunostaining pictures. Yellow arrows indicate nuclei-localized FoxO1 and blue arrows indicate cytosol-localized FoxO1 in cardiomyocytes. *Right:* the results of quantitative analyses (*n* = 6). **f** Myocardial sections of the ischemic border zone from indicated mice with or without I/R were stained with anti-FoxO1 antibody (green), anti-troponin T antibody (red), and DAPI (blue). FoxO1-positive nuclei were counted in the LV tissues. *Upper:* representative pictures of immunostaining in the LV tissues. *Lower:* the results of quantitative analyses (*n* = 3). All experiments were repeated at least three times, with *n* representing biologically independent replicates. *P* values were determined by two-sided unpaired Student's *t* test in (**e**) or one-way ANOVA followed by Tukey's multiple comparison test in (**f**). Data are mean ± SEM. Source data are provided as a Source Data file.

increased phosphorylation of FoxO1 at Ser209 but did not do so when FoxO1-S209A or FoxO1-PR was overexpressed (Supplementary Fig. 3a). Although the PR mutation of FoxO1 at Ser209 alone did not completely abolish Mst1-mediated phosphorylation of FoxO1, the introduction of a PR mutation at all six FoxO1 sites that can be phosphorylated by Mst1 successfully prevented Mst1-mediated phosphorylation of FoxO1, consistent with the results of the mass spectrometric analysis (Supplementary Fig. 3b). To evaluate the functional role of Mst1-induced FoxO1 phosphorylation at Ser209, Ser215, Ser218, Thr228, Ser232, and Ser243 in mediating nuclear localization of FoxO1, adenovirus (Ad-GFP-FoxO1-WT, Ad-Ad-GFP-FoxO1-PR or Ad-GFP-FoxO1-S209A) was transduced into CMs in the presence of Ad-Mst1. CMs were co-stained with anti-troponin T antibody and DAPI. Although Ad-Mst1 induced nuclear localization of FoxO1-WT, nuclear localization of FoxO1-PR was decreased significantly, consistent with the notion that Mst1-induced phosphorylation of FoxO1 at Ser209/Ser215/Ser218/Thr228/Ser232/Ser243 induces nuclear localization of FoxO1 (Fig. 1e; Supplementary Fig. 3c). Similarly, Mst1-induced increases in nuclear localization were attenuated, but not fully abolished, in FoxO1(S209A) (Supplementary Fig. 3c). Additionally, nuclear localization of FoxO1 in CMs was induced in vivo by either cardiac-specific overexpression of Mst1 or myocardial I/R, a cardiac stress known to activate Mst1[25]; however, it was attenuated when adenovirus harboring FoxO1-PR was injected into the heart. As expected, FoxO1 was undetectable in *FoxO1*-cKO mice (Fig. 1f). Nuclear expression of FoxO1 in response to I/R was also decreased when FoxO1(S209A) was expressed in the heart (Supplementary Fig. 4a). These results suggest that Ser209 is involved in Mst1 phosphorylation-mediated FoxO1 nuclear localization.

Collectively, these results indicate that Mst1 physically interacts with and phosphorylates FoxO1 at Ser209/Ser215/Ser218/Thr228/Ser232/Ser243, which in turn induces nuclear translocation of FoxO1 in CMs. Although phosphorylation of FoxO1 at Ser209 plays an important role in mediating the Mst1-induced nuclear translocation of FoxO1, the individual contributions of phosphorylation at Ser215, Ser218, Thr228, Ser232, and Ser243 to the Mst1-induced nuclear translocation of FoxO1 remain to be elucidated.

## Mst1 inhibits the DNA binding of FoxO1

We evaluated the functional consequence of FoxO1 phosphorylation by Mst1 in CMs. First, we investigated the effect of Mst1 on the expression level of FoxO1 in the heart. Immunoblot analysis showed that the protein expression of FoxO1 in the heart was significantly decreased in *Mst1*-KO mice and transgenic mice with cardiac-specific overexpression of DN-Mst1 (Tg-*DN-Mst1*) compared to wild-type (WT) mice (Fig. 2a). Conversely, the amount of FoxO1 protein in the heart was significantly increased in cardiac-specific Mst1-overexpressing (Tg-Mst1) mice compared to in WT mice. There was no significant difference in the FoxO1 mRNA levels of Tg-*DN-Mst1*, *Mst1*-KO, Tg-*Mst1*, and WT mice (Fig. 2b). FoxO proteins in the cytosol are susceptible to degradation by the ubiquitin–proteasome system[29]. We further evaluated the extent of FoxO1 ubiquitination, using the glutathione S transferase (GST)-tandem ubiquitin-binding entity assay. The level of ubiquitinated FoxO1 in the heart was significantly greater in both Tg-*DN-Mst1* and *Mst1*-KO mice than in WT mice. Conversely, there was significantly less ubiquitinated FoxO1 in the heart in Tg-*Mst1* than in WT mice. Similar results were obtained when the level of FoxO1 ubiquitination was evaluated with immunoprecipitation of FoxO1 followed by anti-ubiquitin immunoblots. These results suggest that Mst1 promotes the stabilization of FoxO1 by inhibiting protein degradation through the ubiquitin–proteasome system (Fig. 2c).

We next evaluated the effect of Mst1-mediated phosphorylation upon the transcriptional activity of FoxO1 using reporter gene assays in CMs. Mst1, but not DN-Mst1, dose-dependently suppressed reporter activity driven by three repeats of the insulin-responsive sequence (IRS), a FoxO binding sequence, with a 3 bp spacer (3×IRS) (Fig. 2d). To investigate whether FoxO1 phosphorylated by Mst1 can bind to the DAF-16 binding element (DBE), another FoxO binding sequence, double-stranded DNA pull-down assays were performed with biotin-labeled double-stranded DNA and bacterially expressed recombinant FoxO1 proteins in vitro (Fig. 2e). FoxO1-WT in the absence of Mst1 was pulled down with biotin-labeled double-stranded DNA comprising three repeats of the DBE with a 3 bp spacer (WT-3×DBE) but not with that comprising a 3×DBE mutant (Mt-3×DBE) in test tubes. FoxO1-WT phosphorylated by Mst1 cannot bind to the WT-3×DBE. Similarly, the FoxO1(S209A) mutant was not able to bind to the WT-3×DBE in the presence of Mst1 (Supplementary Fig. 4b). In contrast, the FoxO1-PR mutant was able to bind with the WT-3×DBE even in the presence of Mst1. Like Mst1-phosphorylated FoxO1, the FoxO1 phospho-mimetic (PM) mutant could not bind to the WT-3×DBE. We also evaluated the interaction between the 3×DBE probes and FoxO1 and its mutants using the electrophoretic mobility shift assay (EMSA; Fig. 2f). The specific interaction between 3×DBE and FoxO1 was abolished when FoxO1 was pre-incubated with Mst1 or FoxO1-PM was used. On the other hand, the interaction between 3×DBE and FoxO1-PR was not inhibited even when FoxO1-PR was pre-incubated with Mst1. These results suggest that although Mst1-mediated FoxO1 phosphorylation protects FoxO1 from protein degradation, it attenuates the DNA-binding ability of FoxO1, thereby suppressing the transcriptional activity of FoxO1 on the DBE (Supplementary Fig. 2b). Furthermore, although Mst1-induced phosphorylation of FoxO1 at Ser209 is important for nuclear translocation of FoxO, additional phosphorylation is required for the dissociation of FoxO1 from the DBE.

## Ablation of FoxO1 exacerbates cardiac dysfunction in Tg-Mst1

In order to evaluate the functional significance of the Mst1–FoxO1 interaction and stabilization of FoxO1 in the nucleus in vivo, we evaluated the effect of FoxO1 upon CM apoptosis stimulated by

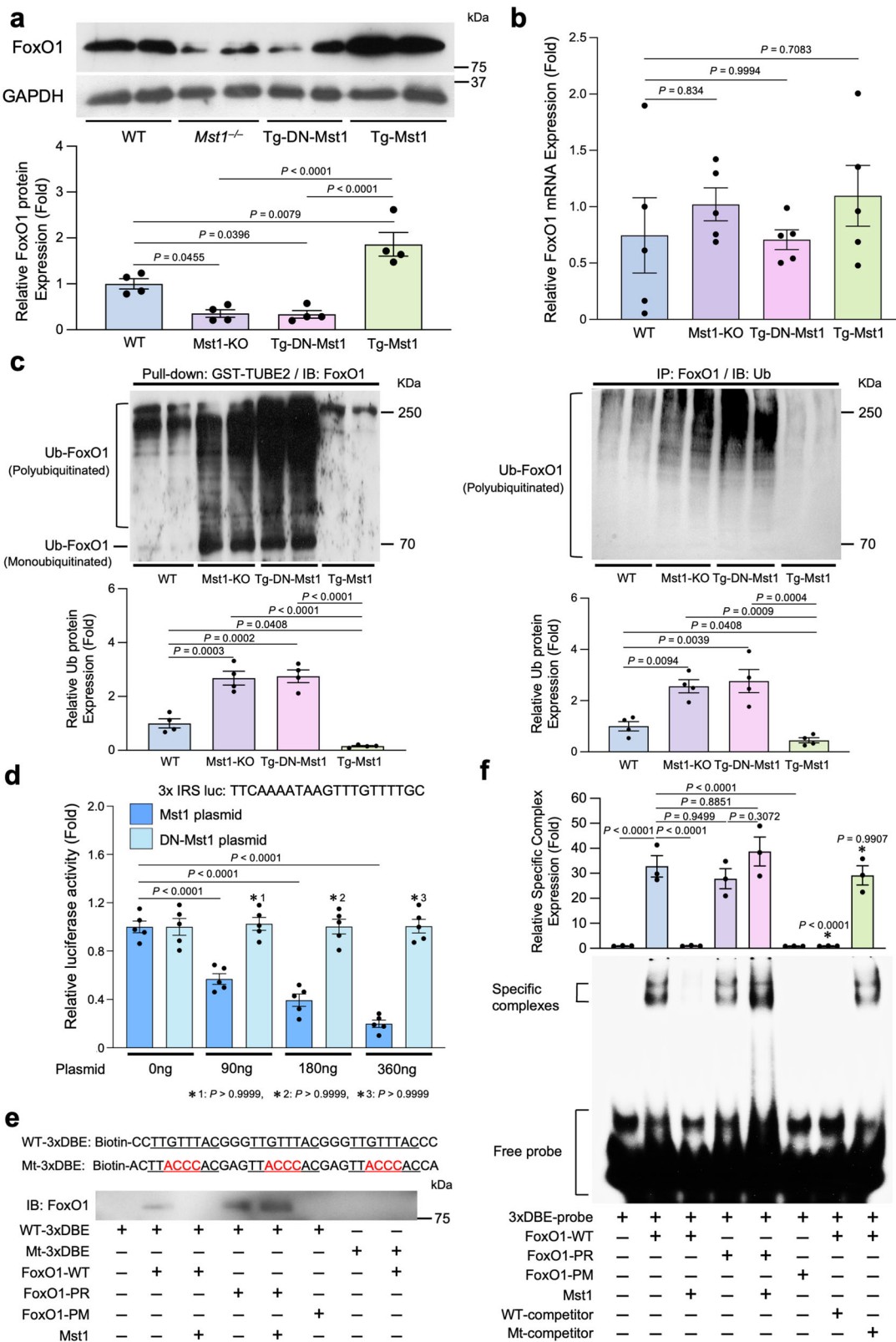

either overexpression of Mst1 or oxidative stress. As reported previously, overexpression of Mst1 alone or wild-type FoxO1 alone increased apoptosis in CMs, as evaluated with TUNEL assays. Mst1-induced increases in CM apoptosis were attenuated in the presence FoxO1 overexpression, whereas they were enhanced when endogenous FoxO1 was downregulated by shRNA knockdown or in the presence of overexpression of FoxO1-PR in vitro (Fig. 3a). Similarly,

$H_2O_2$-induced increases in CM apoptosis were attenuated in the presence of FoxO1 overexpression, whereas they were enhanced when endogenous FoxO1 was downregulated by shRNA knockdown or in the presence of overexpression of FoxO1-PR in vitro (Fig. 3a). These results suggest that although FoxO1 alone promotes apoptosis, FoxO1 may promote cell survival in the presence of Mst1 in cultured CMs.

**Fig. 2 | Mst1 protects FoxO1 from protein degradation but inhibits its transcriptional activity in cardiomyocytes.** **a** Immunoblot analysis of heart homogenates with anti-FoxO1 and anti-GAPDH antibodies. *Upper:* representative images of immunoblot analyses. *Lower:* the results of quantitative analyses ($n = 4$). **b** mRNA expression of FoxO1 in heart homogenates was determined by quantitative RT-PCR ($n = 5$). **c** Heart homogenates were incubated with GST-TUBE2 to obtain ubiquitinated proteins. The samples were then subjected to immunoblot analyses with anti-FoxO1 antibody. *Upper:* representative images of immunoblot analyses. *Lower:* the results of quantitative analyses ($n = 4$). **d** Cardiomyocytes were transfected with a luciferase reporter containing three repeats of insulin-responsive sequence (3×IRS-Luc) and then transfected with pcDNA3.1-Mst1 plasmid at indicated concentrations ($n = 5$). *(1–3) vs. DN-Mst1 plasmid_0 ng of each group. **e** Binding ability of

unmodified and Mst1-phosphorylated recombinant FoxO1-WT to the DAF-16 binding element (DBE) was examined by oligo DNA pull-down assays using biotin-labeled double-stranded DNA with the indicated sequences. Binding ability of recombinant FoxO1 mutants (FoxO1-PR and FoxO1-PM) to DBE was examined as well. **f** Binding ability of unmodified and Mst1-phosphorylated recombinant FoxO1-WT, FoxO1-PR, or FoxO1-PM to DBE was also examined by electrophoretic mobility shift assay. *Upper:* the results of quantitative analyses ($n = 4$). * vs. FoxO1-WT. *Lower:* representative images of autoradiography. All experiments were repeated at least three times, with $n$ representing biologically independent replicates. $P$ values were determined by one-way ANOVA followed by Tukey's multiple comparison test in (**a–d**, **f**). Data are mean ± SEM. Source data are provided as a Source Data file.

We also generated cardiac-specific *FoxO1*-cKO mice by crossing *Myh6*-Cre and *FoxO1*$^{LoxP/LoxP}$ (*FoxO1*$^{L/L}$) mice. We then crossed Tg-*Mst1* mice with *FoxO1*-cKO mice (Tg-*Mst1*-*FoxO1*-cKO). Mst1 overexpression in the heart causes cardiac dysfunction and DCM in mice[25]. Echocardiography conducted at 70–80 days showed that cardiac-specific downregulation of FoxO1 on the background of Tg-*Mst1* significantly decreased left ventricular (LV) ejection fraction and % fractional shortening, indexes of LV systolic function, and increased LV end-diastolic diameter (Fig. 3b, c; Supplementary Table 1). Hemodynamic analyses confirmed that the first derivative of LV pressure (dP/dt) was decreased, while LV end-diastolic pressure was significantly increased, in Tg-*Mst1*-*FoxO1*-cKO mice compared to in Tg-*Mst1* mice (Fig. 3d; Supplementary Table 1). These results suggest that endogenous FoxO1 is protective against cardiac dysfunction induced by Mst1. Postmortem analyses of Tg-*Mst1* at 3 months of age showed dilation of all four cardiac chambers, significant enlargement of both atriums, and reduced wall thickness, as shown previously[25]. Tg-*Mst1*-*FoxO1*-cKO mice exhibited a significant increase in lung weight/tibial length, an index of lung congestion, compared to Tg-*Mst1* mice (Fig. 3e). Although there was no significant difference in the LV myocyte cross-sectional area between Tg-Mst1-*FoxO1*-cKO and Tg-*Mst1*, as determined by wheat germ agglutinin (WGA) staining (Fig. 3f), the level of interstitial fibrosis was significantly increased in the LV of Tg-*Mst1*-*FoxO1*-cKO compared to in Tg-*Mst1*, consistent with the notion that myocyte death and subsequent replacement of the myocardium with fibrous tissue in Tg-*Mst1* mice were exacerbated in the absence of endogenous FoxO1 (Fig. 3g). To evaluate the frequency of cell death in the myocardium, TUNEL staining was performed in these mice. The number of TUNEL-positive CMs was significantly increased in Tg-*Mst1*-*FoxO1*-cKO compared to in Tg-Mst1 mice (Fig. 3H). Taken together, these results suggest that endogenous FoxO1 is a negative feedback regulator of Mst1 in the heart and the CMs therein that serves to prevent Mst1-induced CM apoptosis in vitro and cardiac dysfunction in vivo.

## Co-expression of FoxO1 and Mst1 reduces apoptosis

We next examined the mechanism by which Mst1-induced phosphorylation of FoxO1 inhibits Mst1-induced apoptosis. To this end, mRNA levels of the known FoxO1 target genes were evaluated with RT$^2$ Profiler PCR arrays and individual qRT-PCR analyses in CMs transduced with LacZ, FoxO1, FoxO1 + Mst1, or FoxO1 + DN-Mst1. Analyses of the gene expression profile revealed that known FoxO1 target genes involved in cell death, including *Casp12*, *Fadd*, *Faslg*, *Nox4*, *Pmaip1*, and *Runx2*, are downregulated by overexpression of FoxO1 + Mst1 whereas they are upregulated by overexpression of FoxO1 alone or FoxO1 + DN-Mst1 compared to control (LacZ). On the other hand, antioxidant genes, including, *Cat*, *P2rx7*, *Prdx2*, and *Prdx3*, were upregulated by overexpression of FoxO1 alone or FoxO1 + Mst1 but were all downregulated by overexpression of FoxO1 + DN-Mst1 (Fig. 4a; Supplementary Figs. 5a, b and 6). Consistent with these results, Mst1, but not DN-Mst1, dose-dependently suppressed the activity of luciferase reporters driven by either the *Faslg* promoter or the *Pmaip1* promoter (Fig. 4b).

Conversely, Mst1, but not DN-Mst1, dose-dependently increased the activity of reporter genes driven by the *Cat* promoter or the *Prdx2* promoter in CMs (Fig. 4b). These results suggest that co-expression of Mst1 and FoxO1 upregulates antioxidant genes, whereas it downregulates pro-apoptotic genes.

## Mst1 promotes C/EBP-β-induced transcription of antioxidants

To understand the mechanism by which FoxO1 upregulates antioxidant genes but not those involved in cell death in the presence of Mst1, we analyzed the promoter sequences of both *Faslg* and *Cat*. In particular, we searched for binding sites for major transcription factors present only in the *Cat* promoter but not in the *Faslg* promoter. We noticed that there are three CCAAT boxes and an additional consensus binding site for C/EBP-β in the promoter of *Cat* but not *Faslg* (Supplementary Fig. 7). We next searched transcription factors that bind to the CCAAT box and upregulate *Cat* gene expression and identified C/EBP-β as a possible candidate[20,30]. Furthermore, based on in silico analyses and previous studies conducted on FoxOs, we found that the promoters of some known FoxO target genes, including *Cat* and *Prdx2*, possess C/EBP-β binding site(s) in addition to the FoxO binding site(s), whereas others possess only the FoxO binding site(s) (Supplementary Table 2).

To evaluate the role of FoxO1 and C/EBP-β in mediating Mst1-induced stimulation of *Cat* transcription, we constructed a luciferase reporter driven by the *Cat* promoter with or without mutations in a DBE, three CCAAT boxes, or a potential C/EBP-β binding site (CEBE) (Fig. 4c). Increased expression of Mst1 in CMs led to activation of the *Cat* promoter, and the mutation in the DBE had no significant effect on the Mst1-induced activation of the *Cat* promoter. In contrast, mutations in the CCAAT boxes and/or the CEBE significantly attenuated the Mst1-induced upregulation of the *Cat* promoter activity (Fig. 4c). These results suggest that the CCAAT boxes and/or the CEBE, namely C/EBP-β binding sites, rather than FoxO1 binding sites, are critical for Mst1-mediated activation of the *Cat* promoter. We also evaluated whether C/EBP-β is involved in Mst1-mediated FoxO1 regulation by immunoblotting. Consistent with the qRT-PCR microarray results, co-overexpression of Mst1 and FoxO1 caused a significant increase in catalase and PRDX2 protein levels but a significant decrease in FASLG and PMAIP1 compared to overexpression of FoxO1 alone in CMs (Supplementary Fig. 8a). Knockdown of C/EBP-β by an adenovirus harboring shRNA for C/EBP-β (Ad-sh-C/EBP-β) abolished the increase in catalase and PRDX2 induced by Mst1 co-expressed with FoxO1 (Supplementary Fig. 8a).

We next examined whether Mst1 affects intracellular localization of C/EBP-β in CMs with immunostaining. Endogenous C/EBP-β was localized diffusely in both the cytoplasm and the nucleus. Transduction of Ad-Mst1 promoted nuclear localization of C/EBP-β (Supplementary Fig. 8b). Consistently, immunoblotting of CM lysates after subcellular fractionation demonstrated that overexpression of Mst1 enhanced nuclear localization of C/EBP-β (Supplementary Fig. 8c). The total C/EBP-β level was not significantly altered in *Mst1*-KO, Tg-*DN-Mst1*, or Tg-*Mst1* mouse hearts compared to in control mouse hearts (Supplementary Fig. 8d). Taken together, these results suggest that

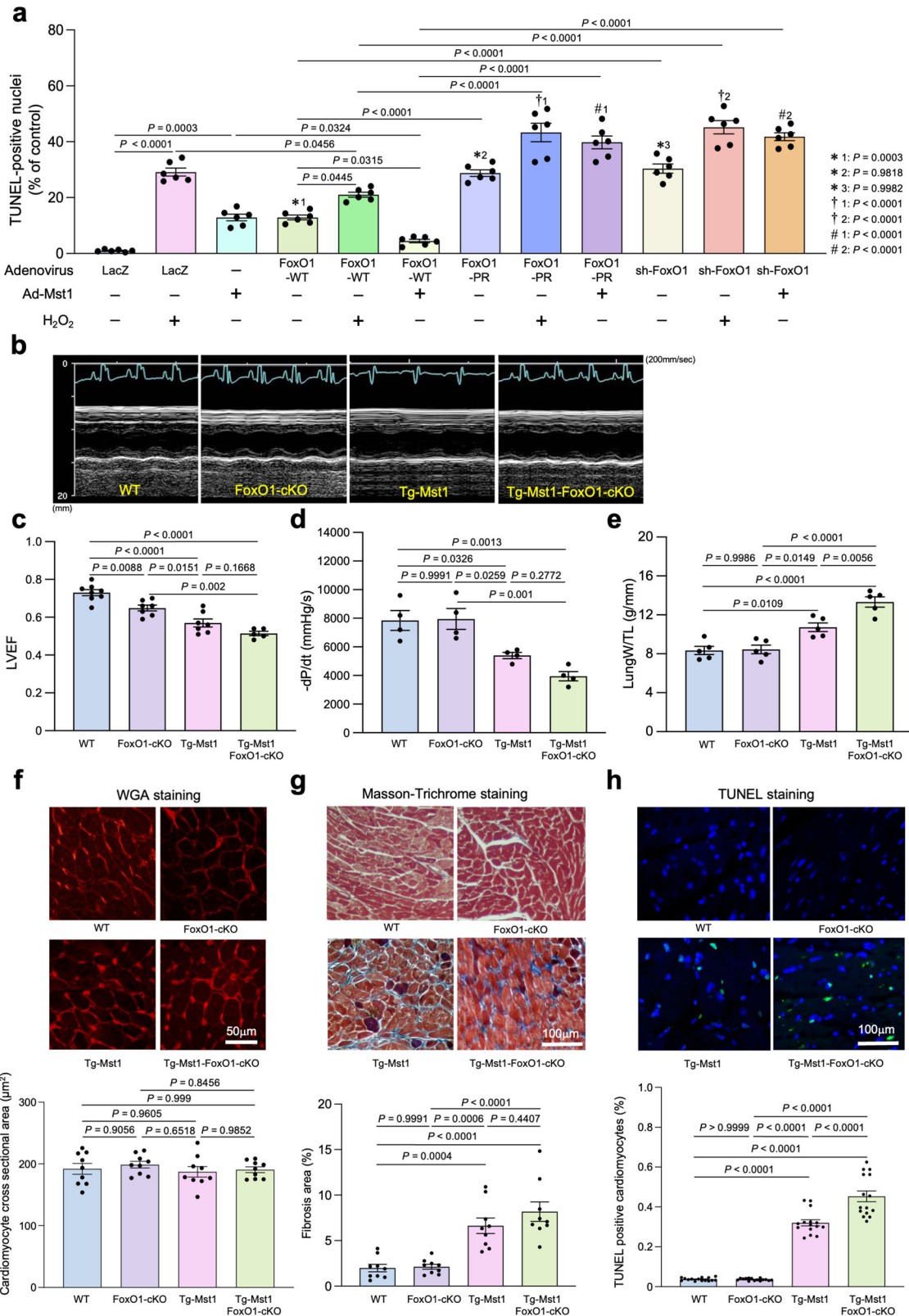

upregulation of antioxidants by Mst1-phosphorylated FoxO1 is mediated through enhancement of C/EBP-β transcriptional activity.

## Mst1 phosphorylates C/EBP-β

We examined whether FoxO1 interacts with C/EBP-β by in vitro pull-down assays. Recombinant full-length human C/EBP-β (C/EBP-β-FL:

amino acid 1–345) and the C-terminal domain of C/EBP-β, which possesses a DBD and a leucine zipper domain (C/EBP-β-CT: 241–345), but not other shorter truncated mutants of C/EBP-β (C/EBP-β-TA: 1–134, C/EBP-β-MID: 135–240), were able to interact with 3×Flag-FoxO1-FL (Fig. 5a), suggesting that FoxO1 directly binds to the C-terminal domain of C/EBP-β.

**Fig. 3 | Phenotypes of *FoxO1*-cKO, Tg-Mst1, and Tg-*Mst1-FoxO1*-cKO bigenic mice.** **a** Quantification of apoptotic cardiomyocytes by counting TUNEL-positive nuclei. The number of TUNEL-positive myocytes was expressed as a percentage of total nuclei, determined by DAPI staining. The experimental data were normalized by the values obtained from control cardiomyocytes without adenovirus transduction ($n = 6$). *(1–3) vs. Ad-LacZ_H$_2$O$_2$(−), †(1–2) vs. Ad-LacZ_H$_2$O$_2$(+), #(1–2) vs. Ad-Mst1. **b** Representative pictures of M-mode echocardiographic imaging in *FoxO1*-cKO, Tg-*Mst1*, Tg-*Mst1* crossed with *FoxO1*-cKO (Tg-*Mst1-FoxO1*-cKO), and WT mice at 70–80 days old. **c** Quantitative analyses of left ventricular ejection fraction in *FoxO1*-cKO ($n = 7$), Tg-*Mst1* ($n = 7$), Tg-*Mst1-FoxO1*-cKO ($n = 5$), and WT ($n = 8$) mice. **d** −dP/dt was measured in *FoxO1*-cKO, Tg-Mst1, Tg-*Mst1-FoxO1*-cKO, and WT mice ($n = 4$). **e** Lung weight/tibia length (TL) was measured in *FoxO1*-cKO, Tg-*Mst1*, Tg-*Mst1-FoxO1*-cKO, and WT mice ($n = 5$). **f** LV myocyte cross-sectional area was measured. *Upper:* LV tissue sections were stained with wheat germ agglutinin−Texas red. *Lower:* the results of quantitative analyses ($n = 9$). **g** Fibrosis area in LV tissues was measured. *Upper:* LV tissue sections were stained by Masson's trichrome staining. *Lower:* the results of quantitative analyses ($n = 9$). **h** TUNEL-positive myocytes were counted among more than 5000 nuclei in the LV tissues of each animal. *Upper:* representative pictures of TUNEL and DAPI staining in the LV tissues. *Lower:* the results of quantitative analyses ($n = 15$). All experiments were repeated at least three times, with $n$ representing biologically independent replicates. $P$ values were determined by one-way ANOVA followed by Tukey's multiple comparison test in (**a**, **c**–**h**). Data are mean ± SEM. Source data are provided as a Source Data file.

We next evaluated whether Mst1 affects the interaction between FoxO1 and C/EBP-β by coimmunoprecipitation assays in cultured CMs. C/EBP-β was found in the FoxO1 immunoprecipitates, and FoxO1 was detected in C/EBP-β immunoprecipitates. In both cases, the interaction between FoxO1 and C/EBP-β was enhanced in the presence of Mst1 overexpression and attenuated in the presence of DN-Mst1 in CMs (Fig. 5b, c). These results suggest that FoxO1−C/EBP-β interaction is promoted by Mst1. Although FoxO1(S209A) physically interacted with C/EBP-β in the presence of Mst1, FoxO1-PR failed to do so in CMs (Supplementary Fig. 8e). Thus, Mst1-induced phosphorylation of serines/threonine other than Ser209 is required for Mst1-induced association between FoxO1 and C/EBP-β.

To determine whether Mst1 phosphorylates C/EBP-β, we conducted in vitro kinase assays using purified Mst1 and GST-C/EBP-β-WT as a substrate. After in vitro kinase reactions, phosphorylation of GST-C/EBP-β-WT was observed in an Mst1-dependent manner (Supplementary Fig. 9a). Thus, Mst1 can directly phosphorylate C/EBP-β. In vitro kinase assays with truncated C/EBP-β proteins as substrates showed that GST-C/EBP-β-CT was phosphorylated by Mst1. However, other C/EBP-β fragments (GST-C/EBP-β-TA and C/EBP-β-MID) were not phosphorylated (Fig. 5d). Mass spectrometric analyses showed that a threonine residue at position 299 (Thr299) in human C/EBP-β is phosphorylated by Mst1 (Fig. 5e; Supplementary Note 2). Thr299 is located in the leucine zipper domain of C/EBP-β and is highly conserved across species (Fig. 5d). To test whether C/EBP-β is phosphorylated at Thr299 in vivo, a phospho-specific antibody against a peptide containing phosphorylated Thr299 was generated. The antibody specifically recognized C/EBP-β-WT but not the C/EBP-β T299A-PR mutant in cultured CMs (Fig. 5f, *upper*). The antibody also recognized endogenous phosphorylated C/EBP-β, and immunoblotting showed increased levels in the presence of Mst1, but not DN-Mst1, in CMs (Fig. 5f, *lower*). Since Thr299 is located in the C-terminal leucine zipper domain of C/EBP-β, we tested whether Mst1-induced phosphorylation affects homodimerization of C/EBP-β. To this end, we expressed Flag-C/EBP-β-WT and HA-C/EBP-β-WT in CMs in the presence or absence of Mst1 and investigated whether Mst1 promotes the interaction between Flag- and HA-tagged C/EBP-β. Interaction between Flag-C/EBP-β-WT and HA-C/EBP-β-WT was enhanced in the presence of Mst1, but attenuated in the presence of DN-Mst1 (Fig. 5g). A Flag-C/EBP-β Thr299Glu PM mutant also interacted with HA-C/EBP-β-PM, even in the absence of Mst1, to the same extent as HA-C/EBP-β-WT in the presence of Mst1, whereas the interaction was not inhibited even in the presence of DN-Mst1 (Fig. 5g). A Flag-C/EBP-β Thr299Ala PR mutant did not interact with HA-C/EBP-β-PM, even in the presence of Mst1 (Fig. 5g). These results suggest that phosphorylation of C/EBP-β at Thr299 by Mst1 facilitates homodimerization of C/EBP-β (Supplementary Fig. 9b). We speculate that the increased homodimerization should enhance the DNA-binding ability of C/EBP-β. Mst1-induced phosphorylation of C/EBP-β at Thr299, as evaluated with a phospho-specific antibody, was completely abrogated in the presence of FoxO1-PR (Fig. 5h), suggesting that FoxO1-PR acts as dominant negative and that phosphorylation and nuclear translocation of FoxO1 by Mst1 are

required for Mst1-induced phosphorylation of C/EBP-β at Thr299. Mst1-induced phosphorylation of C/EBP-β at Thr299 was also abolished in the presence of FoxO1-NLSmut (T24A, S253D, S316A), in which nuclear localization of FoxO1 is abolished, or FoxO1 downregulation (Fig. 5h), further supporting the notion that FoxO1 in the nucleus is required for Mst1-induced phosphorylation of C/EBP-β at Thr299.

Interestingly, FoxO1 interaction with C/EBP-β-PM was induced even in the absence of Mst1 and the suppression of FoxO1−C/EBP-β interaction in the presence of DN-Mst1 was alleviated in the presence of C/EBP-β-PM (Fig. 5c, lanes 5–7). On the other hand, the interaction between FoxO1 and C/EBP-β-PR was inhibited even in the presence of Mst1 (Fig. 5c, lanes 8–10). Thus, although FoxO1 is required for Mst1-induced phosphorylation of C/EBP-β at Thr299, once C/EBP-β is phosphorylated by Mst1, FoxO1−C/EBP-β interaction may take place in an Mst1-independent manner, which may facilitate Mst1-induced phosphorylation.

We evaluated the effect of C/EBP-β phosphorylation at Thr299 upon mRNA expression of known FoxO1 target genes involved in cell death and survival. To this end, CMs were transduced with adenovirus harboring either LacZ (control), C/EBP-β, C/EBP-β-PR, or C/EBP-β-PM, and RT-PCR analyses were conducted. Although C/EBP-β and C/EBP-β-PM enhanced the expression of *Cat* and *Mnsod* compared to LacZ, C/EBP-β-PR did not (Supplementary Fig. 9c). In addition, both C/EBP-β and C/EBP-β-PM decreased *Pmaip1* expression, but C/EBP-β-PR did not (Supplementary Fig. 9c). Overexpression of C/EBP-β increased protein expression of MnSOD, whereas that of C/EBP-β-PM increased protein expression of both MnSOD and catalase compared to overexpression of LacZ (Supplementary Fig. 9d). C/EBP-β-PR did not affect protein levels of either catalase or MnSOD (Supplementary Fig. 9d). Taken together, these results suggest that Thr299 phosphorylation of C/EBP-β positively regulates expression of known FoxO1 target genes that promote cell survival.

## Mst1, FoxO1, and C/EBP-β protect against myocardial ischemia
Hypoxia in cultured rat CMs activated Mst1 and increased phosphorylation of C/EBP-β at Thr251 in vitro, which corresponds to Thr299 in human C/EBP-β (Fig. 6a). Downregulation of Mst1 also abolished the accumulation of C/EBP-β in the nuclei of cultured CMs during hypoxia (Fig. 6b). Knockdown of C/EBP-β further decreased the viability of cultured CMs in response to hypoxia (Supplementary Fig. 10a). Knockdown of Mst1 also exacerbated CM death during short-term hypoxia (within 4 h) (Fig. 6c). Immunoblotting of CM lysates after subcellular fractionation demonstrated that the full-length and a caspase-cleaved form of Mst1 were translocated into the nucleus in response to hypoxia (Fig. 6d), consistent with Mst1 being activated. Thus, the protective effect of Mst1 in response to hypoxia in CMs corresponds to the nuclear accumulation of Mst1.

We then evaluated the role of Mst1, FoxO1, and C/EBP-β during 4 h of myocardial ischemia. The area of myocardial infarction (MI) was significantly smaller in Tg-*Mst1* hearts than in WT hearts. Conversely, suppression of endogenous Mst1 activity in Tg-*DN-Mst1* hearts

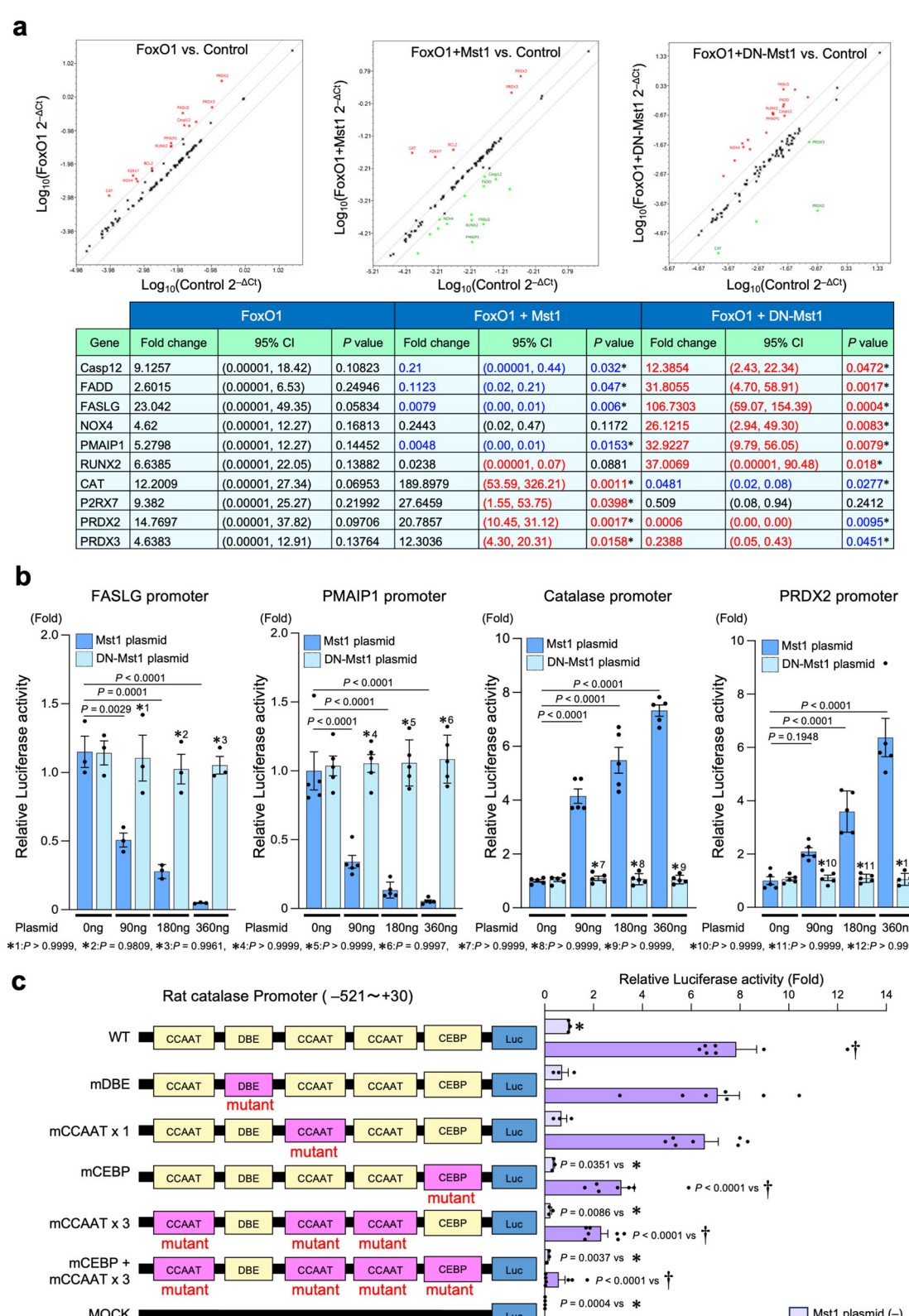

**a**

| Gene | FoxO1 | | | FoxO1 + Mst1 | | | FoxO1 + DN-Mst1 | | |
|---|---|---|---|---|---|---|---|---|---|
| | Fold change | 95% CI | *P* value | Fold change | 95% CI | *P* value | Fold change | 95% CI | *P* value |
| Casp12 | 9.1257 | (0.00001, 18.42) | 0.10823 | 0.21 | (0.00001, 0.44) | 0.032* | 12.3854 | (2.43, 22.34) | 0.0472* |
| FADD | 2.6015 | (0.00001, 6.53) | 0.24946 | 0.1123 | (0.02, 0.21) | 0.047* | 31.8055 | (4.70, 58.91) | 0.0017* |
| FASLG | 23.042 | (0.00001, 49.35) | 0.05834 | 0.0079 | (0.00, 0.01) | 0.006* | 106.7303 | (59.07, 154.39) | 0.0004* |
| NOX4 | 4.62 | (0.00001, 12.27) | 0.16813 | 0.2443 | (0.02, 0.47) | 0.1172 | 26.1215 | (2.94, 49.30) | 0.0083* |
| PMAIP1 | 5.2798 | (0.00001, 12.27) | 0.14452 | 0.0048 | (0.00, 0.01) | 0.0153* | 32.9227 | (9.79, 56.05) | 0.0079* |
| RUNX2 | 6.6385 | (0.00001, 22.05) | 0.13882 | 0.0238 | (0.00001, 0.07) | 0.0881 | 37.0069 | (0.00001, 90.48) | 0.018* |
| CAT | 12.2009 | (0.00001, 27.34) | 0.06953 | 189.8979 | (53.59, 326.21) | 0.0011* | 0.0481 | (0.02, 0.08) | 0.0277* |
| P2RX7 | 9.382 | (0.00001, 25.27) | 0.21992 | 27.6459 | (1.55, 53.75) | 0.0398* | 0.509 | (0.08, 0.94) | 0.2412 |
| PRDX2 | 14.7697 | (0.00001, 37.82) | 0.09706 | 20.7857 | (10.45, 31.12) | 0.0017* | 0.0006 | (0.00, 0.00) | 0.0095* |
| PRDX3 | 4.6383 | (0.00001, 12.91) | 0.13764 | 12.3036 | (4.30, 20.31) | 0.0158* | 0.2388 | (0.05, 0.43) | 0.0451* |

**b**

*1:*P* > 0.9999, *2:*P* = 0.9809, *3:*P* = 0.9961, *4:*P* > 0.9999, *5:*P* > 0.9999, *6:*P* = 0.9997, *7:*P* > 0.9999, *8:*P* > 0.9999, *9:*P* > 0.9999, *10:*P* > 0.9999, *11:*P* > 0.9999, *12:*P* > 0.9999,

**c**

significantly enlarged the area of MI during ischemia. *C/EBP-β*[+/−] and FoxO1-cKO mouse hearts also exhibited greater MI than WT hearts. The suppression of myocardial injury observed in Tg-*Mst1* hearts was reversed significantly when Tg-*Mst1* mice were crossed with *C/EBP-β*[+/−] mice (Fig. 6e). Consistently, the cell-protective effect of Mst1 overexpression was significantly reversed by knockdown of C/EBP-β in cultured CMs (Supplementary Fig. 10b–d). Furthermore, cardiac-specific *C/EBP-β* knockout (*C/EBP-β* cKO) mouse hearts exhibited greater MI than WT hearts following 4 h of ischemia, which was not rescued by FoxO1 overexpression, consistent with the notion that the protective effect of FoxO1 during ischemia is mediated through endogenous C/EBP-β (Supplementary Fig. 11). Collectively, these results suggest that the Mst1−FoxO1−C/EBP-β pathway plays a protective role during ischemia in the heart.

**Fig. 4 | Co-expressed FoxO1 and Mst1 reduce apoptosis of CMs by upregulating antioxidant genes but downregulating pro-apoptotic genes. a** Neonatal rat cardiomyocytes (CMs) were transduced with 10 MOI of adenovirus harboring either LacZ, FoxO1-WT, FoxO1 + Mst1-WT, or FoxO1 + DN-Mst1 in indicated combinations. mRNA expression of FoxO1 target genes was determined by qRT-PCR microarray. *Upper:* representative scatter plots of qRT-PCR microarray analysis of CMs transduced with indicated adenoviruses. FoxO1-associated genes showing increased expression compared to control are shown as red dots. Genes that are downregulated after transduction of indicated adenoviruses compared to control are shown as green dots. *Lower:* quantitative analysis of PCR microarray data. Results are presented as fold change compared to control and were calculated using the ΔΔCt method. 18S rRNA was used as a control. *n* = 3 per group. *P < 0.05 vs. Control group. **b** The effect of Mst1 on the transcriptional activity of FoxO1. Cultured CMs were transfected with reporter genes containing the FASLG promoter, PMAIP1 promoter, catalase promoter, or PRDX2 promoter, together with either pcDNA3.1-Mst1 or pcDNA3.1-DN-Mst1 at indicated concentrations (*n* = 3 for FASLG promoter and *n* = 5 for PMAIP1, catalase, and PRDX2 promoters). * vs. DN-Mst1 plasmid_0 ng of each group. **c** Mutations were introduced into the luciferase reporter construct of the catalase promoter (Supplementary Fig. 7b) that contains the region −521/+30 of the rat catalase gene. The DAF-16 binding element (DBE: ATAAATA) was mutated to AGCCCTA, CCAAT boxes (CCCAT) were mutated to CTGAT, and the C/EBP-β binding element (CEBE: CTCTTGCCTCACG) was mutated to CTCTCAACTCACG. Cardiomyocytes were transfected with the wild type or mutant catalase promoter reporter construct as illustrated in the left panel, with or without 360 ng of pcDNA3.1-Mst1 plasmid (*n* = 3–7/group). * wild type catalase promoter_Mst1 (−), † wild-type catalase promoter_Mst1 (+). All experiments were repeated at least three times, with *n* representing biologically independent replicates. *P* values were determined by two-sided unpaired Student's *t* test in (**a**) or one-way ANOVA followed by Tukey's multiple comparison test in (**b**, **c**). Data are mean ± SEM. Source data are provided as a Source Data file.

### C/EBP-β-T250E knock-in mice are protected against ischemia

To validate the salutary effect of Mst1-induced phosphorylation of C/EBP-β during myocardial ischemia, *C/EBP-β-T250E* knock-in (*C/EBP-β-KI*) mice were generated, using the CRISPR/Cas9 system (Supplementary Fig. 12). Thr250 in mouse C/EBP-β corresponds to Thr299 in human C/EBP-β. Both heterozygous and homozygous *C/EBP-β-KI* mice were born at the expected Mendelian ratio. Thus far, we have characterized the heterozygous mice, and their cardiac function, as evaluated by echocardiographic measurement, was apparently normal at 3 months of age (Supplementary Table 3). After 3 h of myocardial ischemia, the area of MI in *C/EBP-β-KI* mice was significantly smaller than in WT mice (Fig. 7a). Echocardiographic measurements showed that LV dysfunction after myocardial ischemia was attenuated in *C/EBP-β-KI* mice compared to in WT mice (Fig. 7b). These results suggest that phosphorylation of C/EBP-β at Thr250 plays a protective role during myocardial ischemia.

We evaluated the effect of C/EBP-β phosphorylation at Thr250 upon mRNA expression of known FoxO1 target genes involved in cell death and survival at baseline in vivo. mRNA expression of *Cat* and *Mnsod* was significantly greater in *C/EBP-β-KI* mice than in control mice. On the other hand, mRNA expression of *Fasl* and *Bnip3* was significantly lower in *C/EBP-β-KI* mice than in control mice (Fig. 7c). We further conducted ChIP sequencing analyses to evaluate the effect of C/EBP-β phosphorylation at Thr250 (mice) upon the ability of C/EBP-β and FoxO1 to bind to the promoter regions of the *Cat* and *Mnsod* genes during myocardial ischemia in vivo (Fig. 7d). We subjected control and *C/EBP-β-KI* mice to either sham operation or 3 h myocardial ischemia, three mice per group, and ChIP was conducted using chromatin from the three hearts in each group combined. Significant peaks of C/EBP-β binding were observed on the proximal promoters of the *Cat* and *Mnsod* genes. These peaks were larger in *C/EBP-β-KI* mice than in WT mice. These results are consistent with the notion that C/EBP-β regulates transcription of the *Cat* and *Mnsod* genes and that Thr250 phosphorylation of C/EBP-β increases its binding to *Cat* and *Mnsod* (Fig. 7d). FoxO1 also binds to the *Cat* and *Mnsod* promoters where C/EBP-β binds but FoxO1 binding to the *Cat* and *Mnsod* promoters was reduced in the presence of ischemia or in *C/EBP-β-KI*, suggesting that FoxO1 and C/EBP-β bind to the same sites on the *Cat* and *Mnsod* promoters and that their binding may be competitive (Fig. 7d). A similar result was obtained regarding C/EBP-β and FoxO1 binding to the promoter of *Becn1* and *Pink1*, genes involved in autophagy/mitophagy (Supplementary Fig. 13). Whole genome binding of FoxO1 in *C/EBP-β-KI* mice was lower compared to WT regardless of the presence or absence of ischemia, providing further evidence that C/EBP-β and FoxO1 share common binding sites (Supplementary Fig. 14). Motif enrichment analysis showed that C/EBP-β can bind to the FoxO1 binding motif containing the invariable three-middle sequence, GGA. Thus, the dimeric form of C/EBP-β caused by Thr250 phosphorylation can bind to the FoxO1 binding motif with a higher affinity than FoxO1, thereby displacing FoxO1 on the promoter of pro-survival genes (Fig. 7e). ChIP sequencing analyses also showed that FoxO1 binding to the promoter of pro-apoptotic genes, including *Bcl2l11, Fadd, Bnip3, and Pmaip1*, was decreased in the presence of ischemia or in *C/EBP-β-KI*, without concomitant increases in C/EBP-β binding to the promoter (Supplementary Fig. 15). Taken together, these results suggest that Thr250 (mice) phosphorylation of C/EBP-β positively regulates expression of known FoxO1 target genes that promote cell survival in the mouse heart in vivo.

### A phospho-mimetic mutant of C/EBP-β rescues *FoxO1*-cKO mice

Myocardial injury triggered by prolonged ischemia is exacerbated during reperfusion. In order to examine the role of the Mst1−FoxO1−C/EBP-β pathway in mediating protection against myocardial injury during ischemia followed by reperfusion (I/R), another type of ischemic event that promotes myocardial injury, *FoxO1*-cKO mice, cardiac-specific FoxO1 overexpression (Tg-*FoxO1*) mice, and WT mice were subjected to 45 min of ischemia and 24 h of reperfusion. In order to test whether C/EBP-β phosphorylation at Thr250 is sufficient to rescue the increased myocardial injury observed in *FoxO1*-cKO mice, adenovirus harboring the C/EBP-β-Thr299Glu PM mutant (human sequence), 3×Flag-FoxO1-PR, or LacZ was injected into the LV of *FoxO1*-cKO or WT mice 2 days before I/R surgery (Supplementary Fig. 16a). The efficacy of adenovirus injection was confirmed by whole-heart β-galactosidase staining in the mice injected with Ad-LacZ[31]. The size of the area at risk (AAR) did not differ significantly among the mice (Supplementary Fig. 16a, *left graph*). Injection with Ad-LacZ did not significantly affect the size of MI/AAR after I/R compared to control mice without adenovirus injection. Injection with Ad-3×Flag-FoxO1-PR, however, resulted in a significantly larger MI area after I/R than injection with Ad-LacZ (Supplementary Fig. 16a, *middle graph*). The number of TUNEL-positive cells in the ischemic border zone was also greater in Ad-3×Flag-FoxO1-PR-injected mice than in Ad-LacZ-injected mice (Supplementary Fig. 16a, *right graph*). Similarly, the MI area after I/R was significantly larger in *FoxO1*-cKO mice than in WT mice, and the number of TUNEL-positive cells in the ischemic border zone was greater in *FoxO1*-cKO mice than in WT mice. However, injection with Ad-C/EBP-β-PM attenuated the increase in the MI area and the number of TUNEL-positive cells after I/R observed in *FoxO1*-cKO mice. In contrast, the MI area and the number of TUNEL-positive cells were smaller in Tg-*FoxO1* mice than in WT mice (Supplementary Fig. 16a, *right graph*). These results suggest that FoxO1 is protective in the model of I/R, where endogenous Mst1 is activated, and that the supplementation of the PM form of C/EBP-β rescued the increased detrimental effect of I/R caused by the lack of FoxO1. Increased expression of FoxO1 significantly enhanced catalase expression and remarkably decreased $H_2O_2$ release from the myocardium following I/R (Supplementary Fig. 16b). Conversely, deletion of endogenous FoxO1 significantly attenuated catalase expression and remarkably

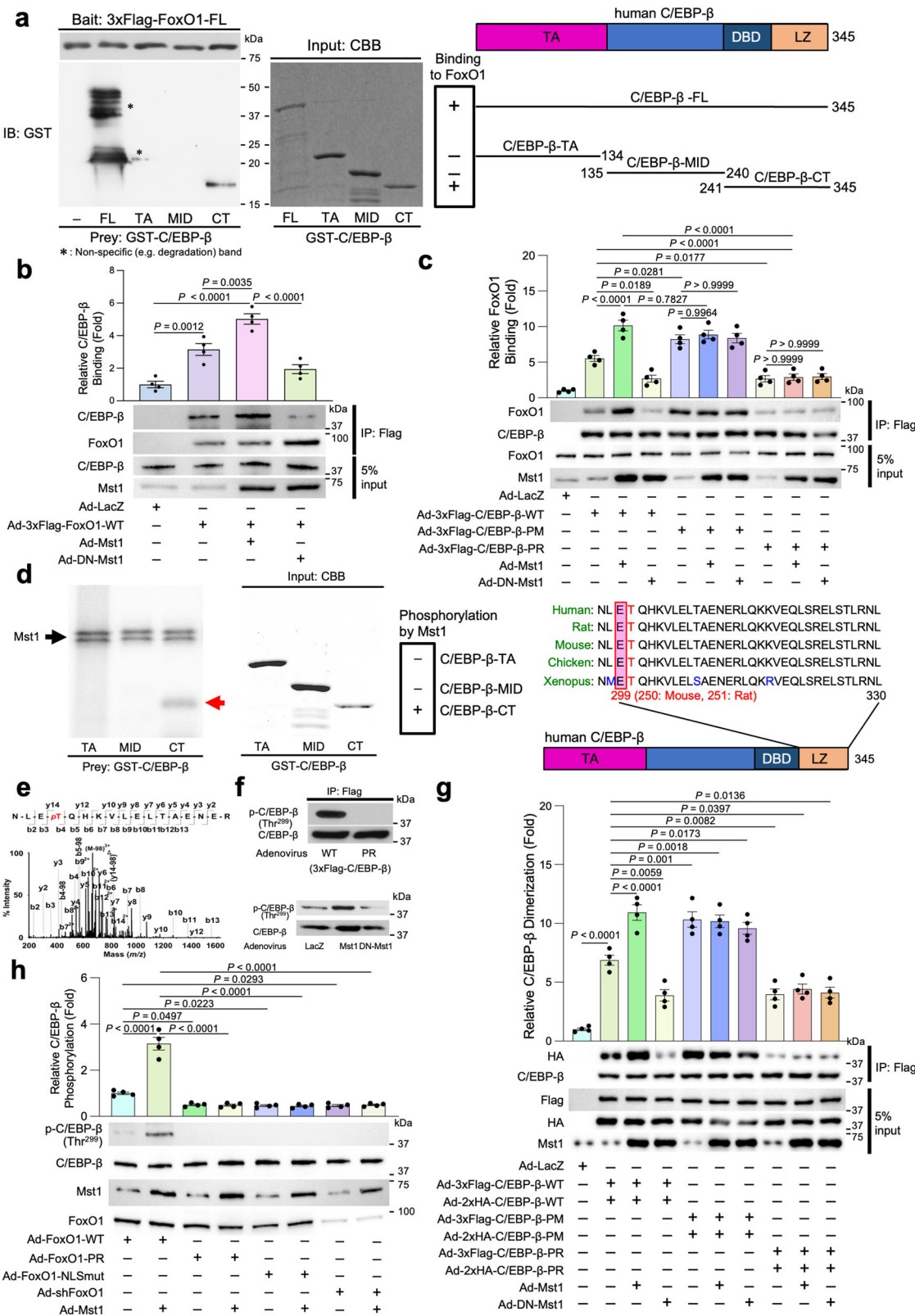

increased $H_2O_2$ release from the myocardium following I/R (Supplementary Fig. 16b). Intramyocardial injection of Ad-3×Flag-C/EBP-β-PM partially rescued the decrease in catalase expression and the increase in $H_2O_2$ release from the myocardium after I/R in *FoxO1*-cKO mice (Supplementary Fig. 16b). Taken together, Thr250-phosphorylated C/EBP-β plays an essential role in mediating the protective effect of endogenous FoxO1 during I/R in the mouse heart in vivo.

## Discussion

Accumulating lines of evidence suggest that FoxO family transcription factors play important roles in development, stress response, apoptosis, and metabolism in the heart. Previous studies have shown that FoxO1 can promote both cell death and survival in the heart[32]. However, how FoxO1 induces transcription of genes with diametrically opposite functions in a context-dependent manner has not been

**Fig. 5 | Mst1 phosphorylates C/EBP-β, thereby promoting the interaction between FoxO1 and C/EBP-β. a** *Left:* interaction between FoxO1 and C/EBP-β was examined by pull-down assays. *Middle:* CBB staining of the gel after SDS–PAGE. *Right:* diagrams of recombinant human C/EBP-β proteins (full-length and partial) used as preys for pull-down assays. DBD: DNA-binding domain, LZ: leucine zipper domain, TA: transactivation domain. **b** Coimmunoprecipitation assays with cardiomyocyte lysates. *Upper:* the results of quantitative analyses (*n* = 4). *Lower:* representative images of immunoblot analyses. **c** Coimmunoprecipitation assays with cardiomyocyte lysates. *Upper:* the results of quantitative analyses (*n* = 4). *Lower:* representative images of immunoblot analyses. **d** *Left:* in vitro kinase assays were carried out by incubating recombinant Mst1 with recombinant C/EBP-β. Reactions were analyzed by SDS–PAGE followed by autoradiography. *Middle:* CBB staining of the recombinant protein-loaded gel after SDS–PAGE. *Right:* alignment of sequences of the leucine zipper domain of human/mouse/rat/chicken/Xenopus C/EBP-β. **e** MS/MS spectrum of a triply charged ion (*m/z* 702.01) corresponding to the phosphopeptide NLE*p*TQHKVLELTAENER from 3xFlag-C/EBP-β phosphorylated by Mst1. The y- and b-ion series confirmed the peptide sequence and

phosphothreonine localization. **f** *Upper:* cardiomyocytes were transduced with Ad-wild-type 3×Flag-tagged C/EBP-β or Ad-3×Flag-tagged C/EBP-β-PR mutant. Ectopically expressed C/EBP-β proteins were immunoprecipitated with an anti-Flag-M2 antibody and detected with an antibody against either C/EBP-β phosphorylated at Thr299 or total C/EBP-β. *Lower:* cardiomyocytes were transduced with Ad-Mst1, Ad-DN-Mst1, or Ad-LacZ. Expression and phosphorylation of C/EBP-β were examined by immunoblots with specific antibodies. **g** An assay for C/EBP-β dimerization. *Upper:* the results of quantitative analyses (*n* = 4). *Lower:* representative images of immunoblot analyses. **h** Cardiomyocytes were transduced with Ad-FoxO1-WT, Ad-FoxO1-PR mutant, or Ad-FoxO1 nuclear localization signal defective mutant (NLSmut) with or without Ad-Mst1. Expression of phospho-C/EBP-β (Thr299), C/EBP-β, Mst1, and FoxO1 was evaluated with immunoblot analyses. *Upper:* the results of quantitative analyses (*n* = 4). *Lower:* Representative images of immunoblot analyses. All experiments were repeated at least three times, with *n* representing biologically independent replicates. *P* values were determined by one-way ANOVA followed by Tukey's multiple comparison test in (**b**, **c**, **g**, **h**). Data are mean ± SEM. Source data are provided as a Source Data file.

described. In this study, we demonstrate that Mst1-induced phosphorylation of FoxO1 shifts the Mst1–FoxO1 complex away from the typical FoxO binding sequences in the promoters of pro-apoptotic genes to C/EBP-β, thereby inducing Thr299 (human)/250 (mouse)/251 (rat) phosphorylation and transcriptional activation of C/EBP-β. Mst1-induced phosphorylation of C/EBP-β at Thr299/250 induces homodimerization of C/EBP-β and activates transcription of pro-survival genes. These results suggest that posttranscriptional modification of FoxO1 switches its bonding partner and induces transcription of a distinct set of genes in CMs.

Oxidative stress induces Mst1-mediated phosphorylation of FoxOs in neuronal cells[14,15]. Activation of Mst1–FoxO signaling can lead to either cell death or cell recovery in response to oxidative stress in a context-dependent manner. Both FoxO1 and FoxO3 phosphorylated by Mst1 induce cell death in neuronal cells. On the other hand, over-expression of *cst-1*, the Mst1 homolog in *C. elegans*, increases lifespan in a *DAF-16*-dependent manner[14]. Moreover, the survival of naïve T cells is critically regulated by the Mst1–FoxO1 signaling pathway[16], genetic defects of which are associated with autosomal recessive primary immunodeficiency[17,18]. We here show that FoxO1 phosphorylated by Mst1 inhibits apoptosis through antioxidant induction and/or inhibition of pro-apoptotic mechanisms in CMs. Cardiac-specific genetic ablation of FoxO1 in Tg-*Mst1* mice exacerbated cardiac dysfunction, indicating that endogenous FoxO1 serves as a powerful negative feedback regulator to prevent Mst1-induced apoptosis in CMs. Co-expression of Mst1 and FoxO1 decreases expression of pro-apoptotic genes, such as *Faslg* and *Pmaip1*, whereas it upregulates expression of antioxidant genes, such as *Cat* and *Prdx2*; thus, FoxO1 selectively stimulates pro-survival mechanisms with the aid of Mst1.

In the current study, we detected that Mst1 phosphorylates the DBD of mouse FoxO1 at Ser209, Ser215, Ser218, Thr228, Ser232, and Ser243 using mass spectrometric analyses (Supplementary Fig. 2a). The crystal structure of the FoxO1-DBD indicates that a region spanning from the α3 helix to the wing2 portion is critical for interaction with the major groove of FoxO binding sequences, such as the IRS and the DBE (Supplementary Fig. 2b)[28]. All of the sites phosphorylated by Mst1 identified in this study (Ser209, Ser215, Ser218, Thr228, Ser232, and Ser243 in mouse FoxO1) or previous investigations (Ser209, Ser215, Ser231, and Ser232 in mouse FoxO1) are located between the α3 helix and the wing2 portion in the FoxO1-DBD. Since phosphorylation of these residues confers a negative charge to the FoxO1-DBD, it should reduce the DNA-binding affinity of FoxO1 through electrostatic repulsion (Supplementary Fig. 2a, b)[28]. Consistent with this notion, previous studies and our results showed that FoxO1 phosphorylated by Mst1 has a diminished DNA-binding affinity[28,33]. Thus, although Mst1 increases the nuclear localization of FoxO1, it decreases the binding of FoxO1 to DNA.

Of the six sites phosphorylated by Mst1 in FoxO1, phosphorylation of Ser209 was found to be essential for the nuclear translocation of FoxO1, consistent with a previous study[15]. However, the Ser209Ala mutation alone failed to prevent the dissociation of FoxO1 from the FoxO1 DBE. Thus, although phosphorylation at Ser209 plays an essential role in mediating Mst1-induced nuclear translocation of FoxO1, it is not sufficient for the dissociation of FoxO1 from the FoxO1 DBE. Phosphorylation of FoxO1 at Ser215, Ser218, Thr228, Ser232, or Ser243 is needed for dissociation of FoxO1 from the DBE and/or association with C/EBP-β. Further investigation is required to identify which of the six serine/threonine phosphorylation sites are required for the Mst1-induced dissociation of FoxO1 from the FoxO1 DBE.

What is the significance of the Mst1-mediated nuclear localization of FoxO1 and dissociation of FoxO1 from the FoxO1 DBE? We here demonstrate that Mst1-induced phosphorylation of FoxO1 in its DBD promotes the binding of FoxO1 to C/EBP-β. Mst1 phosphorylates C/EBP-β in a FoxO1-dependent manner, further increases its interaction with FoxO1, and enhances its transcriptional activity. Accumulating lines of evidence suggest that FoxO can regulate transcription independently of direct DNA binding[34]. Namely, FoxOs associate with various transcription factors, thereby indirectly stimulating or repressing transcription. Our current results suggest that FoxO1 phosphorylated by Mst1 participates in transcriptional regulation through association with other transcription factor(s), such as C/EBP-β. We found that the interaction between FoxO1 and C/EBP-β is direct and that Mst1 enhances their interaction. Since Mst1 can phosphorylate C/EBP-β directly in test tubes, we speculate that FoxO1 acts as either an adapter or a scaffold within the Mst1–FoxO1 complex, thereby bringing Mst1 into the proximity of C/EBP-β to induce phosphorylation. Once C/EBP-β is phosphorylated by Mst1, it can independently stimulate FoxO1–C/EBP-β interaction. It is possible that FoxO1 acts as a co-factor to stimulate the transcription of C/EBP-β. However, we believe this possibility to be remote since expression of C/EBP-β-PM is sufficient to stimulate transcription of *Cat* and *Mnsod* in vivo and in vitro.

We identified Thr299/250 as the major site of phosphorylation by Mst1 in C/EBP-β. The DNA-binding affinity of bZIP transcription factors, including C/EBP-β, is regulated by homo- or heterodimerization of paired leucine zippers[35]. Previous studies demonstrated that phosphorylation of C/EBP-β at Ser272 or Ser276 induces dimerization of the bZIP domain of C/EBP-β[36,37]. As Thr299/250, located in the N-terminus of the bZIP domain, is adjacent to the positively charged amino acid-rich DNA-binding domain, phosphorylation of Thr299/250, which confers a negative charge to Thr299/250, should enhance its interaction with positively charged amino acid(s), thereby promoting dimerization of the bZIP domain.

Among the previously identified targets of FoxO1, both wild type C/EBP-β and C/EBP-β-PM upregulated mRNA expression of pro-

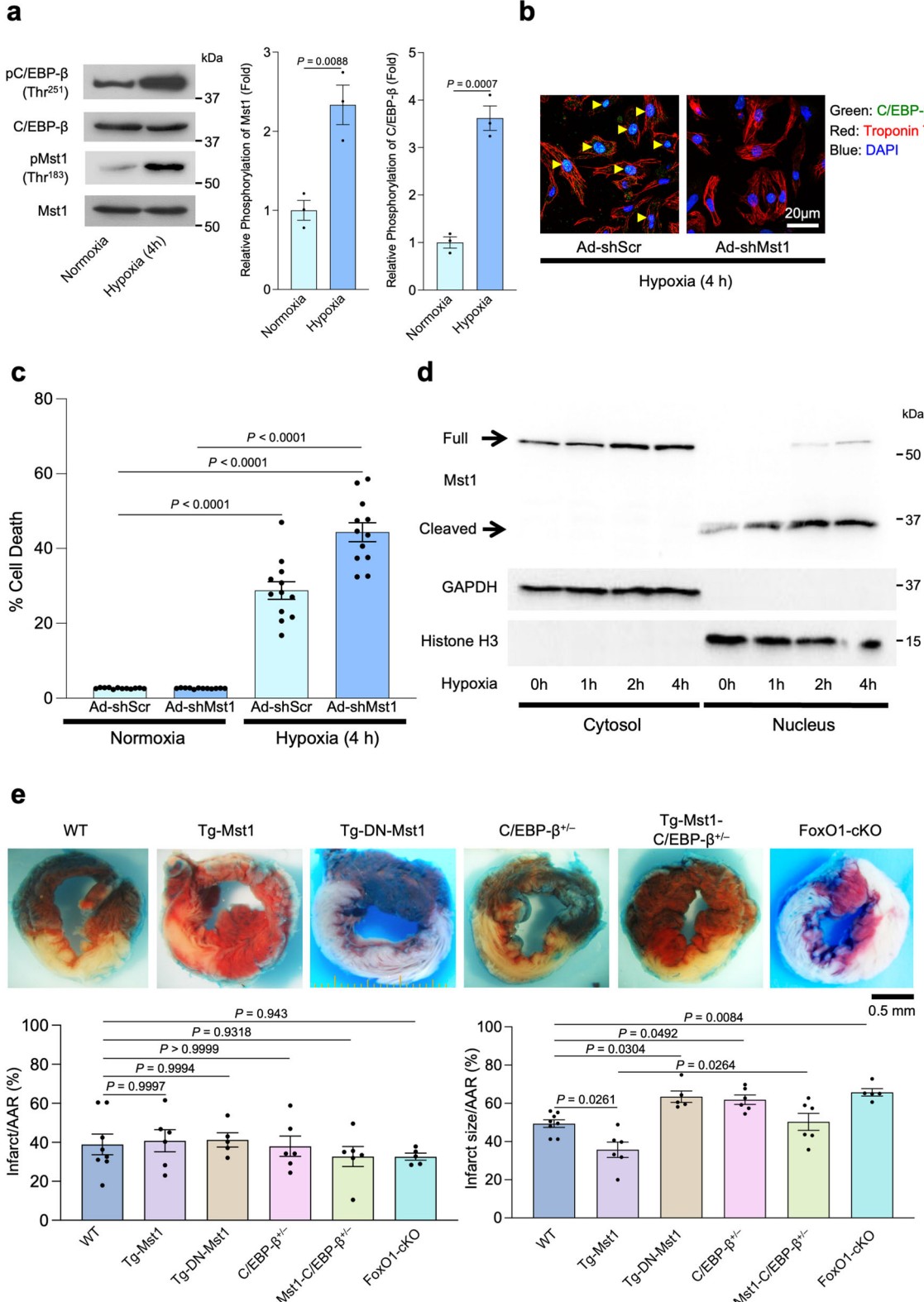

survival genes, including *Cat* and *Mnsod*, but downregulated mRNA expression of *Pmaip1, Faslg*, and *Bnip3* pro-death genes. Whereas the *Cat* and *Mnsod* promoters have binding sites for both FoxO1 and C/EBP-β, those of *Pmaip1, Faslg*, and *Bnip3* only have FoxO1 binding sites. Reporter gene assays showed that the *Cat* promoter is stimulated through the C/EBP-β binding sites rather than the FoxO1 binding sites in the presence of oxidative stress. Together with the results of ChIP

assays, showing that FoxO1 and C/EBP-β bind to the same sites on the *Cat* and *Mnsod* promoters and that their binding may be competitive, these results suggest that Mst1-induced phosphorylation of C/EBP-β at Thr299/250 stimulates transcription of previously identified pro-survival FoxO1 target genes. Although how Thr299/250 PM C/EBP-β inhibits expression of *Pmaip1* and *Faslg* remains to be clarified, we speculate that Mst1-induced phosphorylation C/EBP-β at Thr299/250

**Fig. 6 | Mst1, FoxO1, and C/EBP-β play protective roles in response to prolonged ischemia in the heart. a** Proteins from cultured cardiomyocytes exposed to 4 h of hypoxia were detected with an antibody against either C/EBP-β phosphorylated at Thr299 or Mst1 phosphorylated at Thr183. *Left:* representative images of immunoblot analyses. *Right:* the results of quantitative analyses ($n = 3$). **b** Cardiomyocytes treated with Ad-shMst1 or Ad-shScramble were stained with a C/EBP-β antibody (green), a troponin T antibody (red), and DAPI (blue). **c** Cardiomyocytes were transduced with 10 MOI of adenovirus harboring either shMst1 or shScramble in indicated combinations. Seventy-two hours after the transduction of adenoviruses, some samples were exposed to hypoxia for 4 h, then cardiomyocytes were harvested. The survival of cardiomyocytes was quantitated by CellTiter-Blue assay. The experimental data were normalized by values obtained from control cardiomyocytes without adenovirus transduction ($n = 12$). **d** Proteins in fractionated cellular lysates from cultured cardiomyocytes exposed to indicated periods of hypoxia were detected with Mst1, GAPDH (cytosolic marker), and Histone H3 (nuclear marker) antibodies. **e** Tg-*Mst1* ($n = 6$), Tg-*DN-Mst1* ($n = 5$), *C/EBP-β*$^{+/−}$ ($n = 6$), *C/EBP-β*$^{+/−}$ crossed with Tg-*Mst1* (bigenic) ($n = 6$), *FoxO1*-cKO ($n = 5$), and WT ($n = 8$) mice were subjected to prolonged ischemia for 4 h. *Upper:* gross appearance of LV myocardial sections after Alcian blue and TTC staining. *Lower left graph:* comparison of AAR (percentage of LV) among indicated groups. *Lower right graph:* comparison of infarct size/AAR among indicated groups. All experiments were repeated at least three times, with *n* representing biologically independent replicates. *P* values were determined by two-sided unpaired Student's *t* test in (**a**) or one-way ANOVA followed by Tukey's multiple comparison test in (**b**, **e**). Data are mean ± SEM. Source data are provided as a Source Data file.

may facilitate displacement of FoxO1 from the IRS/DBE through unknown mechanisms.

Mst1 is activated in the heart by I/R and during heart failure and promotes apoptosis of CMs[25,38]. However, we show here that activation of Mst1 stimulates a negative feedback mechanism, consisting of phosphorylation of FoxO1 and C/EBP-β, to promote CM survival. If the level of C/EBP-β Thr299/250 phosphorylation can be increased without activation of Mst1, it may be possible to upregulate cell survival without activating the detrimental functions of Mst1, including activation of the mitochondria-dependent mechanism of apoptosis[5].

Our study has two limitations. First, our study does not distinguish between the various FoxO family proteins. Specifically, we found that Mst1 also facilitates the nuclear translocation of FoxO3 and FoxO4, as well as FoxO1. Although the different isoforms may play significant but redundant roles in mediating the effect of Mst1, we cannot formally exclude the possibility that FoxO3 and/or FoxO4 may regulate the expression of a unique set of antioxidant genes. Another limitation is related to the in vitro experimental system. All of our in vitro experiments were performed with neonatal rat CMs, which may not fully reproduce the signaling mechanism used in adult CMs.

In summary, phosphorylation of FoxO1 by Mst1 attenuates the pro-apoptotic function of FoxO1 by diminishing its DNA-binding ability while stimulating the cell-protective function of FoxO1 by promoting Mst1-mediated upregulation of C/EBP-β transcriptional activity (Supplementary Fig. 17). Mst1-induced phosphorylation of FoxO1 is a negative feedback mechanism that alleviates the detrimental function of Mst1 in the heart.

## Methods
### Mice
All animal protocols were approved by the Institutional Animal Care and Use Committees of the Rutgers New Jersey Medical School (Protocol Number: 201900140) and the Tokyo Medical and Dental University (Permit Numbers: G2018-134C5, G2024-006A, A2023-108C, and A2024-034A), following the Guide for the Care and Use of Laboratory Animals published by the U.S. National Institutes of Health. Handling of mice and euthanasia with $CO_2$ in an appropriate chamber was conducted in accordance with guidelines on euthanasia of the American Veterinary Medical Association. Rutgers is accredited by AAALAC International, in compliance with Animal Welfare Act regulations and Public Health Service (PHS) Policy on Humane Care and Use of Laboratory Animals, and has a PHS Approved Animal Welfare Assurance with the NIH Office of Laboratory Animal Welfare (D16-00098 (A3158-01)). Mice were housed in a temperature and humidity-controlled environment within a range of 21–23 °C with 12-h light/dark cycles, in which they received food and water ad libitum. We used age-matched male mice in all animal experiments.

Male C57BL/6J wild-type mice (Strain #000664) were purchased from Jackson Labs at 5–8 weeks of age, and all genetically modified mice were established in or back-crossed to the C57BL/6J strain. Tg-*Mst1* have been described[25]. Tg-*FoxO1* was generated with the αMHC

promoter (courtesy of J. Robbins, University of Cincinnati, Cincinnati, OH, USA). Cardiac-specific *FoxO1*-cKO mice were generated by mating *FoxO1*$^{L/L}$ mice (courtesy of Ronald Depinho, Harvard Medical School, Boston, MA) with *αMHC-Cre* mice (courtesy of Michael D. Schneider, Imperial College London, London, UK). *C/EBP-β*$^{+/−}$ mice were purchased from Jackson Laboratory (Stock No. 006873, Bar Harbor, ME). Cardiac-specific *C/EBP-β* cKO mice were generated by mating *C/EBP-β*$^{L/L}$ mice with *αMHC-Cre* mice. The *C/EBP-β-KI* mice were established using the CRISPR/Cas9 system (Cyagen Biosciences Inc., Santa Clara, CA) (Supplementary Fig. 12). The mouse *Cebpb* gene (GenBank accession number: NM_00963.4; ENSMUSG00000056501) is located on mouse chromosome 2. As Thr250 is located in exon 1, exon 1 was selected as the target site. The gRNA targeting vector and oligo donor (with targeting sequence, flanked by 120 bp homologous sequences combined on both sides) were designed. The Thr250Glu (ACG to GAG) mutation in the oligo donor was introduced into exon 1 by homology-directed repair. Cas9 mRNA, sgRNA, and oligo donor were co-injected into zygotes for KI mouse production. The pups were then genotyped by PCR followed by sequence analysis. Positive founders were bred to the next generation, which was genotyped by PCR and DNA sequencing analysis. Genetically altered mouse models generated in this study are available from the lead contact upon reasonable request.

### Primary culture of neonatal rat ventricular cardiomyocytes
Primary cultures of ventricular cardiomyocytes were prepared from 1-day-old Crl: (WI) BR-Wistar rats (Harlan Laboratories, Somerville, NJ, USA) as described previously[39]. A cardiac myocyte-rich fraction was obtained by centrifugation through a discontinuous Percoll gradient as described. We obtained myocyte cultures in which more than 95% were myocytes, as assessed by immunofluorescence staining with a mAb against sarcomeric myosin (MF20).

### Antibodies
The following antibodies were used: catalase (ab16731), GFP (9F9.F9) (ab1218), tubulin (ab7291), and troponin T (45932) (Abcam); Mst1 (611052) and MnSOD (611581) (BD Biosciences); FASLG (3330) (BioVision); phospho-Mst1 (Thr183) (3681), FoxO1 (2880), phospho-FoxO1 (Ser256) (9461), 14-3-3α/β (9636), DYKDDDDK (=FLAG) Tag (9A3) (8146), PRDX2 (46855), GAPDH (14C10) (2118), GST (2622), Histone H3 (9715), HRP-linked anti-mouse IgG (7076), and HRP-linked anti-rabbit IgG (7074) (Cell Signaling Technology); FoxO1 (TA323072) and C/EBP-β (TA312911) (Origene); NOXA (=PMAIP1) (sc-26919) and HRP-linked anti-goat IgG (sc-2020) (Santa Cruz Biotechnology); Sarcomeric myosin (MF20) (MAB4470) (R&D Systems); cardiac Troponin T (MA5-12960), phospho-FoxO1 (Ser207) (44-1230G), Alexa Fluor 488-conjugated anti-mouse IgG (A-11029), Alexa Fluor 488-conjugated anti-rabbit IgG (A-11034), Alexa Fluor 594-conjugated anti-mouse IgG (A-11020), and Alexa Fluor 594-conjugated anti-rabbit IgG (A-11037) (Thermo Fisher Scientific). A phosphorylation-specific antibody was raised against a synthetic peptide of the leucine zipper domain of human C/EBP-β, AKMRNLE(T-p)QHKVLELC.

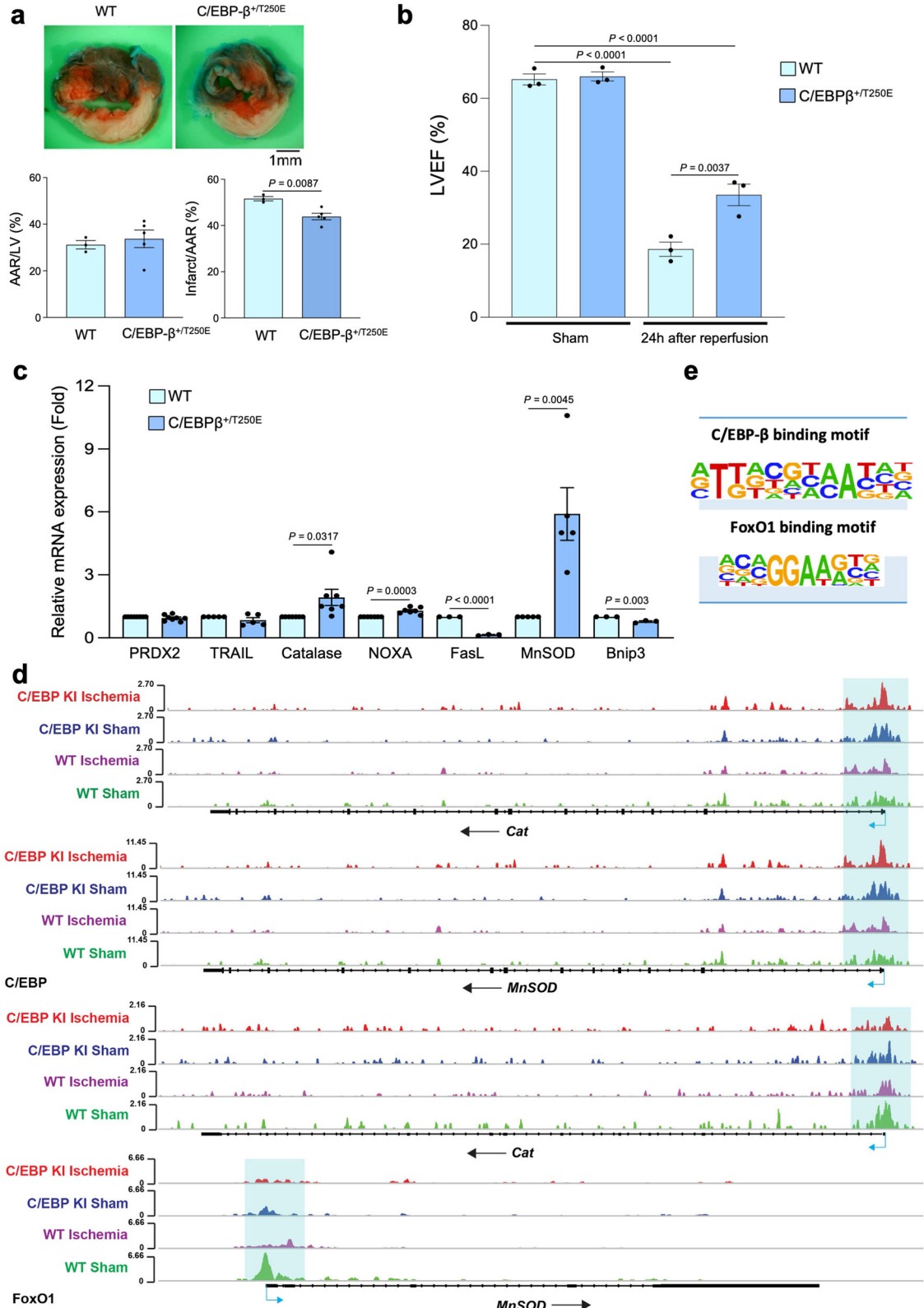

## Construction of adenoviruses

Recombinant adenovirus vectors were constructed, propagated, and titered as previously described[39]. Briefly, pBHGloxΔE1,3 Cre (Microbix), including the ΔE1 adenoviral genome, was co-transfected with the pDC316 shuttle vector containing the gene of interest into 293 cells using Lipofectamine 2000 (Invitrogen). Through homologous recombination, the test genes were integrated into the E1-deleted adenoviral genome. The viruses were propagated in 293 cells. We made replication-defective human adenovirus type 5 (devoid of E1) harboring Mst1, FoxO1-WT, FoxO1-PR mutant, FoxO1-PM mutant, C/EBP-β-WT, C/EBP-β-T299A mutant (C/EBP-β-PR), and C/EBP-β-T299E mutant (C/EBP-β-PM). The FoxO1-PR, FoxO1-PM, C/EBP-β-PR, and C/EBP-β-PM plasmids were generated by site-directed mutagenesis. Adenovirus harboring β-galactosidase (Ad-LacZ) was used as a control.

**Fig. 7 | *C/EBP-β-T250E* knock-in mice are protected against myocardial ischemia. a** *C/EBP-β-T250E* knock-in (*C/EBP-β*-KI) and WT mice were subjected to 45 min of ischemia and 24 h of reperfusion. *Upper:* the gross appearance of LV myocardial sections after Alcian blue and TTC staining. *Lower left graph:* comparison of the size of the AAR (percentage of LV) in the two groups (*n* = 5). *Lower right graph:* comparison of the infarct size/AAR in the two groups (*n* = 5). **b** Quantitative analyses of left ventricular ejection fraction in *C/EBP-β*-KI and WT mice after 45 min of ischemia and 24 h of reperfusion (*n* = 3). **c** qRT-PCR analyses of RNA extracted from heart homogenates of *C/EBP-β*-KI or WT mice to evaluate the effect of C/EBP-β phosphorylation at Thr250 upon mRNA expression of cell death- and survival-associated genes, including PRDX2 (*n* = 8), TRAIL (*n* = 5), catalase (*n* = 7), NOXA (*n* = 7), FasL (*n* = 3), MnSOD (*n* = 5), and Bnip3 (*n* = 3). **d** Chromatin immunoprecipitation assays

with sequencing to examine the effect of C/EBP-β phosphorylation at Thr299 upon the binding ability of C/EBP-β and FoxO1 to the promoter regions of *Cat* and *Mnsod* genes during myocardial ischemia. Upper two panels indicate the binding ability of CEBP-β to the promoter regions of the *Cat* and *Mnsod* genes in WT and *C/EBP-β*-KI ischemic hearts. Lower two panels indicate the binding ability of FoxO1 to the promoter regions of the *Cat* and *Mnsod* genes in WT and *C/EBP-β*-KI ischemic hearts. **e** Motif enrichment analyses showed similarities in binding sequences of C/EBP-β and FoxO1. All experiments were repeated at least three times, with *n* representing biologically independent replicates. *P* values were determined by two-sided unpaired Student's *t* test in (**a**, **c**) or one-way ANOVA followed by Tukey's multiple comparison test in (**b**). Data are mean ± SEM. Source data are provided as a Source Data file.

### Construction of shRNA adenoviral expression vectors
The pSilencer 1.0-U6 expression vector was purchased from Ambion. The U6 RNA polymerase III promoter and the polylinker region were subcloned into the adenoviral shuttle vector pDC311 (Microbix). Targeting sequences of each gene were listed in Supplementary Table 4. Recombinant adenoviruses were generated by homologous recombination in HEK293 cells as described above.

### Gene silencing via siRNA transfection
The siRNAs utilized in this study were pre-designed Silencer Select siRNAs obtained from Thermo Fisher Scientific. The specific IDs for the pre-designed siRNAs, as provided by the manufacturer, are as follows: Rat siC/EBPβ (# 4390771) and siControl (# 4390843). Initially, Lipofectamine RNAiMAX (Thermo Fisher Scientific) was dissolved in Opti-MEM medium. This solution was subsequently combined with the siRNAs at a concentration of 10 nM and incubated for 10–15 min at room temperature. The resulting Lipofectamine–siRNA complex was then introduced into suitable cultured CM plates, which were free of serum and antibiotics, followed by gentle agitation of the mixture. Twenty-four hours after the addition, the media in each culture plate were replaced with DMEM/F-12 supplemented with penicillin/streptomycin antibiotics without serum.

### Recombinant proteins
In order to generate GST-FoxO1 (full-length and fragment), we used pCold-GST vector (Takara/Clontech)[40]. Proteins fused to GST were expressed in *Escherichia coli* strain BL21 (DE3) cultured in LB medium containing 100 μg/ml ampicillin. Protein expression was induced by the addition of 1 mM IPTG followed by a downshift of temperature from 37 to 15 °C. Cells were cultured for 16 h following induction and purified with glutathione Sepharose 4B (GE Healthscience). In order to remove the N-terminal GST tag, the recombinant protein immobilized on glutathione Sepharose 4B was digested at 4 °C with GST-tagged human rhinovirus 3C protease (PreScission Protease, GE Healthcare).

### Immunoprecipitation and immunoblotting
Immunoprecipitation and immunoblotting were carried out as described[39]. Cells were lysed with RIPA buffer (50 mM Tris-HCl (pH 7.4), 0.1% IGEPAL CA-630 0.5% deoxycholate, 10 mM EDTA, 150 mM NaCl, 50 mM NaF, 1 μM leupeptin, and 0.1 μM aprotinin). Lysates were centrifuged at 13,200 rpm for 15 min at 4 °C prior to immunoprecipitation or immunoblotting. The nuclear and cytosolic fractions from myocytes and mouse hearts were prepared with NE-PER Nuclear and Cytoplasmic Extraction Reagents (PIERCE). In immunoprecipitation analyses, samples were incubated with antibodies immobilized to protein A agarose for at least 2 h at 4 °C. After immunoprecipitation, the samples were washed with RIPA buffer three times. They were then eluted with 2× sample buffer and the immunoprecipitates were subjected to immunoblotting.

### Phos-tag SDS–PAGE
The lysates from cultured cardiomyocytes transduced with Ad-FoxO1-WT, Ad-FoxO1-S209A, or Ad-FoxO1-PR, with or without Ad-Mst1, were lysed using RIPA buffer with protease inhibitors (Sigma) and phosphatase inhibitors (Sigma). Samples were loaded onto a 10% polyacrylamide Phos-tag gel (25 μM Phos-Tag [Fuji Film-Wako] and 100 mM MnCl_2). Subsequently, the samples were analyzed by immunoblotting with anti-FoxO1 antibody to detect FoxO1 phosphorylated by Mst1.

### Pull-down binding assays
A GST-fusion protein was mixed in 500 μl of PBS containing 0.5% Triton X-100 and incubated for 30 min at 4 °C. A slurry of glutathione Sepharose 4B with immobilized proteins was added, followed by further incubation for 30 min at 4 °C. After washing three times with PBS, proteins were eluted with 2× SDS sample buffer. The eluates were subjected to SDS–PAGE and stained with Coomassie brilliant blue or immunoblotted.

### In vitro kinase assays
In vitro kinase assays were carried out as described[39]. Recombinant active Mst1 (10 ng) (Millipore) was incubated with full-length or fragmented FoxO1 (1 μg) in kinase buffer (50 mM HEPES pH 7.4, 15 mM MgCl_2 and 200 μM sodium vanadate containing 100 or 5 μM ATP and 10 Ci [γ-$^{32}$P] ATP per reaction) at 30 °C for 30 min. Phosphorylated proteins were separated by SDS–PAGE gel electrophoresis and analyzed by autoradiography.

### Immunofluorescence
Neonatal cardiomyocytes were cultured on coverslips, transduced with 10 MOI of adenoviruses and fixed in 4% paraformaldehyde. Cells were permeabilized with 0.1% Triton X-100 in PBS, and nonspecific binding was blocked with 3% BSA in PBS for 90 min. Overnight incubation with specific antibodies was followed by a 2-h incubation with Alexa Fluor 488 Dye- or Alexa Fluor 594 Dye-conjugated secondary antibody (Invitrogen). Coverslips were washed and mounted on glass slides with a reagent containing DAPI (VECTASHIELD; Vector Laboratories Inc.).

### Isolation of ubiquitinated proteins
Heart homogenate (500 μg) was incubated with Glutathione S transferase-tandem ubiquitin-binding entity 2 (GST-TUBE2, 20 μg, Life Sensors, Malvern, PA) at 4 °C overnight. After adding glutathione sepharose beads, the samples were incubated at 4 °C for 1 h and then washed with RIPA buffer five times. Samples were boiled and subjected to SDS–PAGE analyses.

### Quantitative RT-PCR
Total RNA was prepared from mouse hearts with TRIzol (Invitrogen). Reverse transcription and quantitative PCR were carried out as previously described[38]. Expression values were standardized by those of 18S rRNA. PCR primers of each gene were listed in Supplementary Table 4.

## RT² Profiler PCR array

The expression profile of 91 FoxO1-related genes was determined using a 96-well format custom rat FoxO1 signaling pathway RT² Profiler PCR array (No. CLAR23789, QIAGEN, Hilden, Germany) according to the manufacturer's instructions. The array also included two house-keeping genes and three RNAs as internal controls. qPCR was run on an ABI StepOnePlus qPCR instrument equipped with software version 2.0, using RT² SYBR Green/ROX qPCR master mix (Applied Biosystems, UK). Data analysis was carried out by the $2^{-\Delta\Delta Ct}$ method on the manufacturer's web portal http://pcrdataanalysis.sabiosciences.com/pcr/arrayanalysis.php (QIAGEN, Hilden, Germany).

## Luciferase assay

The rat PMAIP1 promoter, catalase promoter, and PRDX2 promoter were amplified using the PCR primers listed in Supplementary Table 4. The amplified promoters were ligated into pGL3-basic (Promega) to generate luciferase reporter constructs. Mutation of the rat catalase promoter-luc was performed by site-directed mutagenesis as described in Fig. 5. A luciferase reporter driven by the FASLG promoter was obtained from Dr. M. Greenberg at Harvard Medical School, Boston, MA. The 3×IRS-luc luciferase reporter plasmid was obtained from Dr. Kun-Liang Guan at the University of California, San Diego, La Jolla, CA. Measurement of luciferase activity was carried out as described previously[41].

## Double-stranded DNA pull-down assay

Double-stranded DNA pull-down assays were carried out as described previously with modification[41]. Briefly, cells were lysed with a lysis buffer consisting of the following components: 10 mM Tris (pH 8.0), 3 mM CaCl$_2$, 1% IGEPAL CA-630, 320 mM sucrose, 0.1 mM EDTA, 1 mM DTT, and Protease Inhibitor Cocktail (Sigma). Cell lysates were diluted with an equal volume of 2× pull-down buffer containing 20 mM HEPES (pH 8), 100 mM KCl, 5 mM MgCl$_2$, 2 mM EDTA, 2 mM DTT, and 0.2% IGEPAL CA-630. Biotin-labeled sense and non-biotin-labeled antisense oligonucleotides were annealed using a thermal cycler to generate biotin-labeled double-stranded DNA. The biotin-labeled double-stranded DNA (total 15 pmol/100 µl) and 15 µl of streptavidin–agarose (Thermo Fisher Scientific) were added to the lysate and gently rotated at 4 °C for 2 h. The streptavidin–agarose was then washed three times with 1 ml of 1× pull-down buffer and eluted with 2× sample buffer, and the eluate was subjected to immunoblotting.

## Electrophoretic mobility shift assay (EMSA)

A double-stranded oligonucleotide probe containing the DBE was end-labeled with [γ-³²P] ATP. The labeled probe was incubated with 50 ng of FoxO1-WT recombinant protein with or without Mst1-mediated kinase reaction or FoxO1 mutant proteins (FoxO1-PR, FoxO1-PM) in 20 µl of the reaction mixture (20 mM Tris-HCl, pH 8.0, 40 mM KCl, 5% glycerol, 0.4 mM DTT, 0.2 mM EDTA, 2 mM MgCl$_2$, 1 mg/ml BSA, and 20 ng of poly(dI·dC)). After incubation for 15 min on ice, the reaction mixtures were directly loaded onto a 6% polyacrylamide gel and electrophoresed in 0.5× TBE (44.5 mM Tris, 44.5 mM boric acid, 1 mM EDTA, pH 8.3). The gels were dried and analyzed by autoradiography.

## Chromatin immunoprecipitation (ChIP) assay

ChIP assays were performed using the SimpleChIP Enzymatic Chromatin IP kit (Cell Signaling Technology) with several modifications, as described previously[41]. To measure the collected chromatin fragments, quantitative PCRs were carried out using the oligonucleotide primers indicated in Supplementary Table 4.

## Chromatin immunoprecipitation sequencing (ChIP-seq)

ChIP-seq was performed according to the ActiveMotif® protocol. CMs harvested from WT and C/EBP-β mice were crosslinked with 1% formaldehyde for 15 min, following glycerin incubation for 5 min at room temperature. Cells were washed with cold PBS twice, lysed, and sonicated to shear DNA. C/EBP-β and FoxO1 antibodies were used to immunoprecipitate protein/chromatin complexes. Several washing steps were followed by protein digestion using proteinase K. Reverse crosslinking was carried out at 65 °C. DNA was subsequently purified. ChIP-seq library preparation was performed with TruSeq ChIP Library Preparation Kit (Illumina, San Diego, CA, USA). Sequencing was performed on an Illumina NextSeq500. Bowtie was used to align sequence reads. ChIP-seq package from Bioconductor was used for peak calling and batch annotation. Homer was used for motif enrichment. USC IGV browser was used for peak visualization.

## Mass spectrometry

We performed a kinase reaction using either recombinant GST-tagged mouse FoxO1-DBD or Flag-tagged human C/EBP-β, together with recombinant active Mst1 protein ($n = 1$ in each experiment). Both unphosphorylated and phosphorylated proteins were separated by SDS–PAGE and stained with Coomassie brilliant blue. The gel bands were excised into ~1 cm³ pieces and washed four times with 1 mL of a solution containing 30% acetonitrile (ACN) and 70% 100 mM NH$_4$HCO$_3$. Next, 200 µL of 25 mM dithiothreitol (DTT) solution was used to reduce disulfides at 55 °C for 30 min. This was followed by the addition of 200 µL of 50 mM iodoacetamide solution to alkylate thiols at room temperature in the dark for 30 min. The gel pieces were then dehydrated with 200 µL of ACN to remove DTT and iodoacetamide. For in-gel trypsin digestion, we added 100 µL of trypsin solution (4 µg/mL in 50 mM NH$_4$HCO$_3$) to each sample and incubated them at 37 °C overnight. The resulting peptides were extracted, desalted using Pierce C$_{18}$ spin columns (Thermo Scientific) according to the manufacturer's protocol, and concentrated with a SpeedVac before LC–MS/MS analysis on an Orbitrap Velos tandem MS instrument (Thermo Scientific). Peptides from each sample were reconstituted in Solvent A (2% ACN in 0.1% formic acid (FA)). Two microliters of peptides from each sample were analyzed by LC–MS/MS using an Orbitrap Velos Mass Spectrometer coupled with an UltiMate 3000 nano LC system (Thermo Scientific). Peptide separation was performed on an Acclaim PepMap C$_{18}$ column (75 µm × 15 cm, 3 µm, 100 Å) using a 2-h binary gradient from 2% to 95% of Solvent B (85% ACN in 0.1% FA), at a flow rate of 300 nL/min. Eluted peptides were introduced to the MS system via a Nanospray Flex ion source with a spray voltage of 2 kV and a capillary temperature of 275 °C. MS spectra were acquired in positive ion mode in a data-dependent acquisition manner. The lock mass was used for accurate mass measurements. Full MS scans were performed in an m/z range of 375–1500 in profile mode, with an AGC value of ~1E6. Following each full MS scan, the top 15 intensity peptides with charge states from $2^+$ to $7^+$ were selected with an isolation window of 2 m/z for MS/MS analysis using collision-introduced dissociation. The dynamic exclusion time was 45 s. The MS/MS spectra were searched against a Uniprot mouse or human database using both the Mascot and Sequest search engines on the Proteome Discoverer platform (PD V1.4). Trypsin was specified as the enzyme with two missed cleavages. Methionine oxidation and STY phosphorylation were set as variable modifications, while C carbamidomethylation was a fixed modification. Peptide mass tolerance was set to 10 ppm, and MS/MS mass tolerance was set to 0.5 Da. The false discovery rate for both protein and peptide identification was below 1%. To estimate phosphorylation site localization probability, the PhosphoRS node in PD software was used. Only phosphorylation sites with probabilities of 90% or higher were considered confidently assigned.

## Measurement of H$_2$O$_2$ concentration

H$_2$O$_2$ levels were determined using an Amplex Red® H$_2$O$_2$ assay kit (Invitrogen) as described previously[41].

## Echocardiography

Mice were anesthetized using 12 μl/g body weight of 2.5% tri-bromoethanol (Avertin, Sigma), and echocardiography was performed as described previously[25] with a 13-MHz linear ultrasound transducer. Two-dimensional guided M-mode measurements of LV internal diameter were obtained from at least three beats and then averaged. LV end-diastolic dimension (LVEDD) was measured at the time of the apparent maximal LV diastolic dimension, and LV end-systolic dimension (LVESD) was measured at the time of the most anterior systolic excursion of the posterior wall. LVEF was calculated using the following formula: LVEF (%) = $100 \times (LVEDD^3 - LVESD^3)/LVEDD^3$.

## Hemodynamic measurements

Mice were anesthetized with 2.5% Avertin (0.29 mg/kg i.p.). A 1.4-French Millar catheter was then inserted through the right carotid artery into the ascending aorta and advanced into the LV, where pressures and the first derivative of LV pressure over time (d$P$/dt) were recorded. TAC was conducted as described previously[31].

## Histological analyses

Heart specimens were fixed with formalin, embedded in paraffin, and sectioned at 6-μm thickness. Interstitial fibrosis was evaluated by Masson-Trichrome staining as described[31]. Myocyte cross-sectional area was measured from images captured from WGA-stained 1-μm-thick sections as described[31]. Suitable cross sections were defined as having nearly circular capillary profiles and circular-to-oval myocyte sections. No correction for oblique sectioning was made. The outline of 100–200 myocytes was traced in each section. Adobe Photoshop software (Creative Suites 3 Extended Version; Adobe Systems Inc., Mountain View, CA, USA) was used to determine myocyte cross-sectional area.

## Evaluation of apoptosis in tissue sections

DNA fragmentation was detected in situ using TUNEL, as described[25]. Briefly, deparaffinized sections were incubated with proteinase K, and DNA fragments were labeled with fluorescein-conjugated dUTP using TdT (Roche Molecular Biochemicals). Nuclear density was determined by manual counting of DAPI-stained nuclei in six fields for each animal using the 40× objective, and the number of TUNEL-positive nuclei was counted by examining the entire section using the same power objective. Limiting the counting of total nuclei and the TUNEL-positive nuclei to areas with a true cross-section of myocytes made it possible to selectively count only those nuclei that clearly were within myocytes.

## I/R surgery in vivo

Mice were anesthetized by intraperitoneal injection of pentobarbital sodium (60 mg/kg). A rodent ventilator (Model 683; Harvard Apparatus Inc., Holliston, MA, USA) was used with 65% oxygen during the surgical procedure. The animals were kept warm using heat lamps and heating pads. Rectal temperature was monitored and maintained between 36.8 and 37.2 °C. The chest was opened by a horizontal incision through the muscle between the ribs (third intercostal space). Ischemia was achieved by ligating the anterior descending branch of the left coronary artery (LAD) using an 8-0 nylon suture, with silicon tubing (1 mm OD) placed on top of the LAD, 2 mm below the border between the left atrium and LV. Regional ischemia was confirmed by ECG change (ST elevation). After occlusion for 20 min, the silicon tubing was removed to achieve reperfusion. The chest wall was closed with 8-0 silk. The animal was removed from the ventilator and kept warm in a cage maintained at 37 °C overnight. Hearts were harvested after 24 h of reperfusion.

## Intramyocardial gene transfer

Adenoviral vectors containing the FoxO1 S209A/S215A/S218A/T228A/S232A/S243A mutant gene, or β-galactosidase were injected intramuscularly using a syringe with a 30-gauge needle into the LV free wall ($1 \times 10^9$ opu/30 μl)[31]. I/R surgery was performed 3 days after injection. Adeno-associated virus serotype 9 (AAV9) vectors carrying a cardiac troponin T promoter and a FoxO1 Ser209Ala mutant (AAV9-cTnT-FoxO1_S209A) constructed by VectorBuilder Inc. were injected into the jugular vein. I/R surgery was performed 2 weeks after injection.

## Assessment of area at risk and infarct size

After I/R, the animals were re-anesthetized and intubated, and the chest was opened. After arresting the heart at the diastolic phase by KCl injection, the ascending aorta was cannulated and perfused with saline to wash out blood. The LAD was occluded with the same suture, which had been left at the site of the ligation. To demarcate the ischemic AAR, Alcian blue dye (1%) was perfused into the aorta and coronary arteries. Hearts were excised, and LVs were sliced into 1-mm cross sections. The heart sections were then incubated with a 1% triphenyltetrazolium chloride solution at 37 °C for 10 min. The infarct area (pale), the AAR (not blue), and the total LV area from both sides of each section were measured using Adobe Photoshop software, and the values obtained were averaged. The weight of each section was measured using a balance. The percentage of area of infarction and AAR of each section were multiplied by the weight of the section and then totaled from all sections. AAR/LV and infarct area/AAR were expressed as percentages. There was no significant difference in AAR/LV among genetically modified mice and nontransgenic controls.

## Statistics & reproducibility

All statistical analyses were performed using PRISM version 9 (GraphPad Software, San Diego, CA, USA). The sample size required was determined according to a power analysis based on previous studies examining the effects of myocardial ischemia or cardiomyocyte apoptosis. All values are expressed as mean ± SEM. Statistical analyses between groups were performed by an unpaired Student's $t$ test for two groups or one-way analysis of variance (ANOVA) followed by the post-hoc Tukey's method for multiple pairwise comparisons for three groups or more unless otherwise stated. In all cases, the results were considered statistically significant at a $P$ value < 0.05. The statistical analyses used for each figure are indicated in the corresponding figure legends. All experiments are represented by multiple biological replicates or independent experiments. The number of replicates per experiment is indicated in the legends. All experiments were conducted using at least two independent sets of experimental materials or cohorts to reproduce similar results. No sample was excluded from the analysis.

## Reporting summary

Further information on research design is available in the Nature Portfolio Reporting Summary linked to this article.

## Data availability

The results of the microarray and ChIP sequencing analyses are available from the Gene Expression Omnibus via accession codes GSE213010 and GSE213011. All data generated or analyzed during this study, including the mass spectrometry data, are included in this published article and its Supplementary Information. Source data are provided with this paper.

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

## Acknowledgements

We thank Daniela K. Zablocki for critical reading of the manuscript. This work was supported in part by U.S. Public Health Service Grants HL67724, HL91469, HL102738, HL112330, HL138720, HL144626, HL150881, and AG27211, the Foundation Leducq Transatlantic Network of Excellence 15CBD04 (J.S.), American Heart Association Scientist Development Grant 12SDG12070262 (Y.M.), Merit Award 20 MERIT35120374 (J.S.), JSPS KAKENHI Grant-in-Aid for Scientific Research (C) 26461126, 15H04817, 17K09570, and 20K08399 (Y.M.), and the Banyu Fellowship Program sponsored by Banyu Life Science Foundation International (Y.M.).

## Author contributions

Conceptualization: Y.M. and J.S.; methodology: Y.M., J.N., P.Z., and T.L.; formal analysis: Y.M., J.N., and T.L.; investigation: Y.M., J.N., Z.A., P.Z., E.-A.S., K.T., S.M., and T.L.; resources: J.S.; writing original draft: Y.M., J.N., and J.S.; writing—review & editing: J.S.; supervision: T.S., H.L., and J.S.; project administration: J.S.; funding acquisition: Y.M. and J.S.

## Competing interests

The authors declare no competing interests.
