## [Peer Review File · Nature Communications]

REVIEWER COMMENTS

Reviewer #1 (Remarks to the Author):

The present manuscript highlights the role of Hippo signaling kinase Mst in regulating opposite functions (proapoptotic and cell protective) of FoxO in cardiomyocytes. In vivo and in vitro experiments have shown that FoxO regulates both cell death-promoting and -inhibiting mechanisms, even in the same cells. However, the molecular mechanism of these diametrically opposite functions of FoxO is not well established. Authors hypothesized that post-translational modification of FoxOs alters the pairing mechanisms between FoxOs and their target genes, thereby regulating different sets of genes in a context-dependent manner. Authors demonstrated that although FoxO1 promotes transcription of proapoptotic genes, Mst1 converts FoxO1 to a cell-protective molecule through activation of C/EBP- β , thereby protecting the heart against ischemia.

This is a very well-done study. Some minor comments are listed below to improve the manuscript.

In figure 3A, the Addition of Mst1 in the presence of shFoxO1 reduced TUNEL-positive cells compared to shFoxO1 alone. So, in in vivo conditions, genetic deletion of c-FoxO1 in Tg-Mst1 mice should also show reduced apoptosis and improved cardiac function(LVEF). However, Fig 3C and H show increased apoptosis (3H) associated with reduced LVEF in TgMst1;c-FoxO1^{-/-} mice compared to TgMst1 or c-FoxO1^{-/-} mice (3C).

In Sup Figure 5C, C/EBP- β signals are very strong. Does overexpression of Mst1 only enhance the nuclear localization of C/EBP- β or also the stability of the protein by phosphorylating it (similar to FoxO1)? Authors should analyze C/EBP- β protein expression levels in the hearts from WT, Mst1^{-/-}, Tg-DN-Mst1, and Tg-Mst1.

Authors demonstrate that suppression of myocardial injury observed in Tg-Mst1 hearts was reversed significantly when Tg-Mst1 mice were crossed with C/EBP- β ^{+/-} mice. Do changes in myocardial injury also translated to cardiac functions? Please comment or include data if available.

Similarly, were any functional changes observed in C/EBP- β -T250E knock-in (C/EBP- β -KI) after ischemia compared to the controls, consistent with the reduced infarct size? Please comment or include data if available.

Reviewer #2 (Remarks to the Author):

These authors, utilizing a series of gene loss- and gain-of-function strategies *in vitro* and *in vivo*, combined with molecular, immuno-histochemical (IHC), echocardiographic and histological techniques, set out to decipher mechanisms whereby Mst1-mediated phosphorylation of FoxO1 and C/EBP- β governs their nuclear translocation to attenuate oxidative myocardial damage by differentially modulating expression of pro-survival vs pro-death genes in cardiomyocytes. They report a novel and previously unknown cooperative action of Mst1-FoxO1-C/EBP- β axis in cardiomyocyte survival following ischemic insult. However, the precise mechanisms of action remain incompletely understood. Furthermore, the majority of the data related to Mst1-mediated regulation of FoxO activity as well as the role of FoxO in ischemic heart disease have been reported previously.

Fox example, a growing body of literature has already described how Mst1-mediated phosphorylation of FoxO1 and FoxO3 at specific sites governs their DNA binding activity, cellular localization, and interaction with 14-3-3 to modulate cell survival, specific gene expression, angiogenesis, and life span (Yuan Z et al., 2009; Lehtinen MK et al., 2006; Brent MM et al., 2008 and Kim YH et al., 2019). The role FoxO proteins in ischemia-induced myocardial damage has also been reported (Sengupta A et al., 2011). Therefore, much of the data presented here related to the Mst1-FoxO1 axis and FoxO1's role in ischemia-induced myocardial damage lack novelty. Also, some other data are not convincing due to lack of critical control experiments.

Major Points:

1. In Figures 1 and 2, the authors have generated mutant versions of FoxO1 to assess their interaction with and phosphorylation by Mst1. However, most of the data shown in Figures 1, 2, S1 and S2 have already been published (Yuan Z et al., 2009; Lehtinen MK et al., 2006; Brent MM et al., 2008). Besides the four known serine residues in mouse FoxO1 (Ser209, Ser215, Ser218, and Ser225), the authors have identified 2 additional Mst1 phosphorylation sites (Thr228 and Ser243) (Fig. S1C). However, the physiological significance and validation of their phosphorylation *in vitro* or *in vivo* was not shown. More importantly, it has been shown that mutation of Ser212 (mouse Ser209) of human FoxO1 (Yuan Z et al., 2009) or the corresponding serine residue in FoxO3 alone (Lehtinen MK et al., 2006) is sufficient to completely abrogate Mst1-mediated phosphorylation of FoxO1 and FoxO3, respectively. As such, the authors need to use a S209A mutant FoxO1 and test FoxO1 phosphorylation in cardiomyocytes. In addition, control experiments using FoxO1-cKO mice will be important with the *in vivo* data shown in Fig. 1B and F, respectively. Furthermore, data using IP of FoxO1 followed by immunoblot of Ub are essential to support the conclusion that Mst1 protects FoxO1 degradation (Fig. 2C). Moreover, experiments using mutated FoxO1 binding sites or

knockdown of FoxO1 or an Mst1-DN expression plasmid would be required to probe specificity of the data shown in Fig. 2D. Finally, it is not clear how the DNA-binding activity of FoxO1-PR was increased by Mst1 (Fig. 2F).

2. In Figure 3A, it is not clear how cell death (%) for the corresponding transduction conditions varies with or without H₂O₂ treatment. Also the authors need to explain how FoxO1-PR as well as both LOF and GOF of FoxO1 induce apoptosis to the relatively same extent with or without H₂O₂ treatment.

3. In Figure 4B, experiments using Mst1-DM are required to probe the specificity of Mst1-mediated reciprocal regulation of promoter activity. Figure 4C suggests that the FoxO1 binding site is not required for Mst1-dependent rat catalase promoter activity, while Fig. S5B indicates that FoxO1 is required. The authors need to explain why and show whether those transcription factor binding sites are evolutionarily conserved.

4. Based on Figure 5G and Fig. S5D, the authors concluded that phosphorylation and nuclear translocation of FoxO1 by Mst1 are required for Mst1-induced phosphorylation of C/EBP- β (page 17, bottom). Abrogation of such Mst1-mediated C/EBP- β phosphorylation in FoxO1-KD cells or in the presence of an NLS mutant FoxO1 will be required to confirm this conclusion. In addition, using a WT or phosphor-mimic C/EBP- β , the authors stated that Mst1-mediated phosphorylation of FoxO1 and C/EBP- β promotes formation of a trimeric (FoxO1-C/EBP- β -Mst1) complex (Fig. 5B) and C/EBP- β homodimer (Fig. 5F). To support these bold conclusions, experiments using an Mst1-DN and C/EBP- β -PR construct will be necessary in panel B and F, respectively. Homodimer formation of C/EBP- β -PM should be tested in the absence of Mst1.

5. In Figure 7, using a novel C/EBP- β -KI mouse model, the authors have demonstrated the importance of C/EBP- β phosphorylation to attenuate ischemia/reperfusion-induced myocardial damage (Fig. 7A). In Figures 6 and Fig. S12, the authors have also utilized different transgenic and knockout mouse models and demonstrated that loss of Mst1 or FoxO1 activity trigger severe myocardial damage following prolonged ischemia (Fig. 6E) and injection of phosphor-mimic C/EBP- β virus into the LV tissue of FoxO1-cKO mice attenuated ischemic damage (Fig. S12A-B). To strengthen the novelty of these findings reciprocal experiments using C/EBP- β -cKO mice (Fig. S7B) and injection of FoxO1 virus will be essential. In addition, unlike FoxO1-PR, it is not clear why overexpression of FoxO1-WT in cardiomyocytes in vitro (Fig. 3A) and in vivo (Fig. S12) had dramatic inverse effects on cell death.

Minor Points:

1. The figure legend and data are different for Figure 4B.

2. Data for FASLG and PMAIP1 are missing in Fig. S5B.

Reviewer #3 (Remarks to the Author):

Major comments

1. In the results section- titled Mst1 physically interacts with and phosphorylates FoxO1, thereby enhancing nuclear translocation of FoxO1 in CMs, paragraph ending with “Mst1-overexpressing CMs, FoxO1 failed to interact with 14-3-3 proteins”. The authors focus on the role of MST1 in regulating FoxO1 binding to 14-3-3. Du X et al., (J Immunol 2014) have already demonstrated that MST1 stabilizes both FoxO1 and 3 by inhibiting Akt activity. Please comment on the endogenous basal Akt levels and phospho-Akt levels under MST1 activation? Is the inability of 14-3-3 to interact with FoxO1 an Akt mediated effect?

2. The results section entitled - Mst1 protects FoxO1 from degradation but inhibits its DNA binding in CMs, ending in page 11 with sentence – “The results suggest that although Mst1-mediated FoxO1 phosphorylation protects FoxO1 from protein degradation, it attenuates the DNA binding ability of FoxO1, thereby suppressing the transcriptional activity of FoxO1 on the DBE”. The focus again has been completely on MST1. It could be possible that MST1 interaction with FoxO1 DBD could be responsible for preventing transcriptional activity but the prevention of FoxO1 ubiquitination and degradation could be via MST1 mediated inhibition of Akt. The endogenous Akt 1 total and phosphorylation status along with the activity should be measured.

3. In the results section titled – “Co-expressed FoxO1 and Mst1 significantly reduced H2O2-induced apoptosis in CMs by upregulating antioxidant genes and downregulating pro-apoptotic genes”, the authors claim that MST1 regulation of FoxO1 is responsible for upregulating anti-oxidant genes and downregulating pro-apoptotic genes. If MST1 binding prevents the DBD- FoxO1 to bind to its responsive elements, how can they turn on the anti-oxidant genes?

4. If the anti-oxidant genes are a result of CEBP - beta activation, then the MST1-FoxO1 nexus becomes redundant rather finding the direct link between activation of CEBP-beta by MST1 would be a better strategy. In the FoxO1 knock down cells, and in cardiac FoxO1 knock out animals, test whether MST1 activation could lead to CEBP - beta activation and enhancement of transcription function with resultant upregulation of antioxidant genes.

5. There is a high level of divergence in the models used. The authors have moved from using H2O2 to increase oxidative stress and activate MST1 and in latter figures have employed hypoxia model and ischemic model (in vivo) for confirming CEBP- beta’s role in protection. The continuity is lost in translation and it is important to provide evidence whether MST1 is the master regulator of CEBP-

beta in all of the models used. The authors also need to explain whether this CEBP-beta regulation is FoxO1 dependent or independent?

6. It could also be possible that other FoxO family members playing a role in anti-oxidant genes upregulation as shown in Figure-4. Measure the total and nuclear fractions and DNA-binding abilities of FoxO3, and FoxO4 in the presence of MST1 and DN-MST1 conditions.

7. Under conditions of hypoxia or ischemia (both in vitro and in vivo) conditions, a major transcription factor that could be playing a parallel role is HIF1 alpha. Under the given conditions, it is important to measure HIF1 alpha expression in total and nuclear compartments? Also, it is important to measure whether CEBP-beta is complexing in the nucleus with HIF1 alpha along with FoxO1 and other FoxO members.

Minor comments:

1. In the introduction part, second page first paragraph – states stimulate physical interaction between Beclin1 and the Bcl-2 family proteins, thereby inhibiting autophagy. Which Bcl2 family member is being alluded to?

2. In the introduction part- second page second paragraph states that - on the other hand, several other kinases can promote FoxO activation by facilitating the nuclear translocation of FoxO. The authors have loosely mentioned several kinases- need to specify which kinases are being discussed. Also, when the authors mention FoxO activation and translocation- which FoxO family member are they referring to?

3. Neonatal cardiomyocytes being used in this study may perform a few divisions as compared to adult myocytes that are terminally differentiated and do not undergo cell-division. Please comment.

Point by point responses

We thank the reviewers for their careful assessment and constructive criticisms.

Reviewer #1 (Remarks to the Author):

1. In figure 3A, the Addition of *Mst1* in the presence of *shFoxO1* reduced TUNEL-positive cells compared to *shFoxO1* alone. So, in *in vivo* conditions, genetic deletion of *c-FoxO1* in *Tg-Mst1* mice should also show reduced apoptosis and improved cardiac function (LVEF). However, Fig 3C and H show increased apoptosis (3H) associated with reduced LVEF in *TgMst1;c-FoxO1*^{-/-} mice compared to *TgMst1* or *c-FoxO1*^{-/-} mice (3C).

This is a valid point. We repeated the TUNEL experiments and reassessed the results (Figure 3A). The data is now consistent with the *in vivo* data. We speculate that a condition that largely increases cell death could have made the quantification inaccurate. However, we wish to point out that in both *in vitro* and *in vivo* experiments, induction of cardiomyocyte apoptosis in response to *Mst1* overexpression was exacerbated when endogenous *FoxO1* was downregulated. *Mst1* stimulates both apoptosis and survival, the latter of which is mediated through *FoxO1*. Thus, *Mst1* exacerbates cell death in the presence of *FoxO1* downregulation.

2. In Sup Figure 5C, *C/EBP-β* signals are very strong. Does overexpression of *Mst1* only enhance the nuclear localization of *C/EBP-β* or also the stability of the protein by phosphorylating it (similar to *FoxO1*)? Authors should analyze *C/EBP-β* protein expression levels in the hearts from WT, *Mst1*^{-/-}, *Tg-DN-Mst1*, and *Tg-Mst1*.

This is a wonderful suggestion. We evaluated the protein levels of *C/EBP-β* in the hearts of WT, *Mst1*^{-/-}, *Tg-DN-Mst1* and *Tg-Mst1* mice (New Figure S8D) and found that there are no significant differences among the indicated groups.

3. Authors demonstrate that suppression of myocardial injury observed in *Tg-Mst1* hearts was reversed significantly when *Tg-Mst1* mice were crossed with *C/EBP-β*^{+/-} mice. Do changes in myocardial injury also translated to cardiac functions? Please comment or include data if available.

The reviewer raised an important issue. Unfortunately, we recently lost *Tg-Mst1* (line 28), a high *Mst1* expression line. Although we were planning to conduct IVF using frozen sperm available from our collaborator in China, the transfer and the IVF procedure have been delayed and have not yet been completed. In addition, further generating the cross with cardiac specific *C/EBP-β*^{+/-} mice requires a significant amount of time.

In order to show that changes in the size of MI (evaluated with TTC staining) are translated into changes

in cardiac function, we have thus far been able to conduct I/R experiments with cardiac specific *C/EBP-β* +/- mice. Echocardiography experiments conducted 7 days after reperfusion following 45 min of ischemia showed that cardiac specific *C/EBP-β* +/- mice have reduced cardiac function, consistent with the greater infarction size observed 24 hours after I/R (See above). Thus, the increased myocardial injury in cardiac specific *C/EBP-β* +/- mice is translated into reduced cardiac function one week after injury.

4. Similarly, were any functional changes observed in *C/EBP-β-T250E* knock-in (*C/EBP-β-KI*) after ischemia compared to the controls, consistent with the reduced infarct size? Please comment or include data if available.

We conducted new experiments in which we applied 3 hours of ischemia in WT and *C/EBP-β-KI* mice and then evaluated cardiac function with echocardiographic measurements (New **Figure 7B**). We now show that *C/EBP-β-KI* mice exhibited better cardiac function than WT mice, consistent with the MI size results.

Reviewer #2 (Remarks to the Author):

They report a novel and previously unknown cooperative action of Mst1-FoxO1-C/EBP-β axis in cardiomyocyte survival following ischemic insult. However, the precise mechanisms of action remain incompletely understood. Furthermore, the majority of the data related to Mst1-mediated regulation of FoxO activity as well as the role of FoxO in ischemic heart disease have been reported previously.

Fox example, a growing body of literature has already described how Mst1-mediated phosphorylation of FoxO1 and FoxO3 at specific sites governs their DNA binding activity, cellular localization, and interaction with 14-3-3 to modulate cell survival, specific gene expression, angiogenesis, and life span (Yuan Z et al., 2009; Lehtinen MK et al., 2006; Brent MM et al., 2008 and Kim YH et al., 2019). The role FoxO proteins in ischemia-induced myocardial damage has also been reported (Sengupta A et al., 2011). Therefore, much of the data presented here related to the Mst1-FoxO1 axis and FoxO1's role in ischemia-induced myocardial damage lack novelty. Also, some other data are not convincing due to lack of critical control experiments.

Thank you for your comments. The reviewer correctly pointed out previous findings regarding the effect of phosphorylation of FoxO by Mst1 and the role of FoxO1 in myocardial ischemia. We respectfully point out that our study clarifies the *downstream* mechanism by which Mst1-induced phosphorylation of FoxO1 leads to activation of cell survival genes and protects the heart against ischemia, which we believe has not been shown previously.

We agree with the reviewer's comment regarding the control experiments. In this revision, we have conducted additional experiments and included control experiments. We believe that the quality of the data has been improved.

Major Points:

1. *In Figures 1 and 2, the authors have generated mutant versions of FoxO1 to assess their interaction with and phosphorylation by Mst1. However, most of the data shown in Figures 1, 2, S1 and S2 have already been published (Yuan Z et al., 2009; Lehtinen MK et al., 2006; Brent*

MM et al., 2008). Besides the four known serine residues in mouse FoxO1 (Ser209, Ser215, Ser218, and Ser225), the authors have identified 2 additional Mst1 phosphorylation sites (Thr228 and Ser243) (Fig. S1C). However, the physiological significance and validation of their phosphorylation *in vitro* or *in vivo* was not shown. More importantly, it has been shown that mutation of Ser212 (mouse Ser209) of human FoxO1 (Yuan Z et al., 2009) or the corresponding serine residue in FoxO3 alone (Lehtinen MK et al., 2006) is sufficient to completely abrogate Mst1-mediated phosphorylation of FoxO1 and FoxO3, respectively. As such, the authors need to use a S209A mutant FoxO1 and test FoxO1 phosphorylation in cardiomyocytes.

To address the reviewer's concerns, we expressed FoxO1(S209A) in cardiomyocytes in the presence or absence of Mst1. We then evaluated phosphorylation of FoxO1 using Phos-tag SDS-PAGE blotting with a FoxO1 antibody. We found that the mutation at Ser209 alone allowed more phosphorylation by Mst1 than mutation at all phosphorylation sites (New **Figure S3B**). Thus, Ser209 is not the only site in FoxO1 phosphorylated in the presence of Mst1, consistent with the results of the mass spectrometry analysis.

Immunohistochemical analyses indicated that the Mst1-induced nuclear translocation of GFP-FoxO1 in the heart is significantly attenuated (if not completely abrogated) when GFP-FoxO1(S209A) is transduced instead of GFP-FoxO1 (wild type, WT) (New **Figure S3C**). Thus, Ser209 phosphorylation plays an important role in the nuclear localization of FoxO1 in the presence of Mst1. Similar results were obtained with mice subjected to either sham operation or I/R after transduction with either AAV9-cTNT-FoxO1(WT) or AAV9-cTNT-FoxO1(S209A) (**Figure S4A**).

We also evaluated the DNA binding ability of the FoxO1(S209A) mutant using double-stranded DNA pull-down assays *in vitro*. Interestingly, the Mst1-induced suppression of wild type FoxO1 DNA binding to the 3xDBE sequence was not affected when Ser209 alone was mutated to Ala (New **Figure S4B**), whereas the Mst1-induced suppression of FoxO1 binding to the 3xDBE sequence was completely abrogated when all six serine/threonine Mst1 phosphorylation sites in FoxO1 were mutated to Ala (FoxO1-PR). Thus, Ser209 phosphorylation is not required for FoxO1 to dissociate from the conventional FoxO1 binding site in the presence of Mst1. Rather, other serine/threonine residues at Mst1 phosphorylation sites other than Ser209 are required for the FoxO1 dissociation.

Furthermore, immunoprecipitation assays were performed to evaluate the ability of C/EBP- β to bind to the FoxO1(S209A) or FoxO1-PR mutants. Although the S209A mutation did not affect the interaction of FoxO1 with C/EBP- β , the FoxO1-PR mutant was unable to interact with C/EBP- β even in the presence of Mst1 (New **Figure S8E**). Thus, based on these results, we conclude that *although Ser209 phosphorylation is important for Mst1-induced nuclear translocation of FoxO1, other Mst1 phosphorylation sites in FoxO1, besides Ser209, play an important role in mediating dissociation of FoxO1 from its conventional DNA binding site and consequent interaction between FoxO1 and C/EBP- β .*

*In addition, control experiments using FoxO1-cKO mice will be important with the *in vivo* data shown in Fig. 1B and F, respectively.*

We applied transverse aortic constriction (TAC) for 2 weeks in FoxO1-cKO mice and evaluated Mst1-FoxO1 interaction (**Figure 1B**). We also applied ischemia/reperfusion in FoxO1-cKO mice and evaluated staining of FoxO1 (**Figure 1F**). Both experiments serve as negative controls.

Furthermore, data using IP of FoxO1 followed by immunoblot of Ub are essential to support the conclusion that Mst1 protects FoxO1 degradation (Fig. 2C).

We conducted additional experiments and the data is now presented in **Figure 2C**. The results show that FoxO1 is protected from ubiquitination in the presence of Mst1.

Moreover, experiments using mutated FoxO1 binding sites or knockdown of FoxO1 or an Mst1-DN expression plasmid would be required to probe specificity of the data shown in Fig. 2D.

We conducted additional experiments and the data with DN-Mst1 has been added in **Figure 2D**. The results confirmed that Mst1 inhibits the 3xIRS-Luc activity in a dose-dependent manner compared to DN-Mst1.

Finally, it is not clear how the DNA-binding activity of FoxO1-PR was increased by Mst1 (Fig. 2F).

The reviewer raised an interesting question. We speculate that FoxO1 phosphorylation around the DNA binding domain increases the negative charge and reduces its affinity for DNA. Thus, the phosphorylation-resistant mutant may have an increased binding affinity.

2. In Figure 3A, it is not clear how cell death (%) for the corresponding transduction conditions varies with or without H₂O₂ treatment. Also the authors need to explain how FoxO1-PR as well as both LOF and GOF of FoxO1 induce apoptosis to the relatively same extent with or without H₂O₂ treatment.

The reviewer raised an important issue and we now show data *with and without* H₂O₂ (**Figure 3A**). H₂O₂ has a strong pro-apoptotic effect in cardiomyocytes but H₂O₂-induced apoptosis was attenuated in the presence of FoxO1 wild type (WT). On the other hand, the apoptotic effect of H₂O₂ was exacerbated in the presence FoxO1-PR or shFoxO1 due to the lack of stimulation of the anti-apoptotic effects of C/EBP- β .

FoxO1-WT stimulates both pro-apoptotic genes and anti-apoptotic genes at baseline. The net effect could be stimulation of either apoptosis or survival, and the manifestation of the overall phenotype is context-dependent. Importantly, the DNA binding domain of FoxO1-WT is phosphorylated by Mst1 and FoxO1 binding is shifted from the FoxO element (many of them are found in proapoptotic genes) to C/EBP- β (stimulating anti-apoptotic genes). Thus, apoptosis is inhibited in the presence of Mst1.

On the other hand, due to the decreased nuclear entry, shFoxO1 and FoxO1-PR fail to stimulate either pro-apoptotic or anti-apoptotic genes and the net effect could be either stimulation or inhibition of apoptosis. Importantly, *increased apoptosis caused by shFoxO1 or FoxO1-PR cannot be reversed by Mst1*.

3. In Figure 4B, experiments using Mst1-DM are required to probe the specificity of Mst1-mediated reciprocal regulation of promoter activity.

We conducted additional experiments and the data obtained with DN-Mst1 has been added in **Figure 4B**. The reciprocal regulation of Mst1 upon pro- and anti- apoptotic gene promoters was abolished in the presence of DN-Mst1, confirming the specificity of Mst1.

Figure 4C suggests that the FoxO1 binding site is not required for Mst1-dependent rat catalase promoter activity, while Fig. S5B indicates that FoxO1 is required. The authors need to explain why and show whether those transcription factor binding sites are evolutionarily conserved.

The reviewer raised an important issue. When FoxO1 is phosphorylated by Mst1, it functions as an adaptor for Mst1-induced C/EBP- β phosphorylation. C/EBP- β is phosphorylated as a result of the coordinated actions of Mst1 and FoxO1, and FoxO1 can stimulate catalase transcription through C/EBP- β on the catalase promoter. Using ChIP-sequencing analyses, *we show that FoxO1 and phosphorylated C/EBP- β compete for binding to the FoxO1 binding DNA sequence in the catalase and MnSOD promoters.* As shown in **Figure S7**, the catalase gene promoter sequence is conserved in rats, mice and humans.

4. Based on Figure 5G and Fig. S5D, the authors concluded that phosphorylation and nuclear translocation of FoxO1 by Mst1 are required for Mst1-induced phosphorylation of C/EBP- β (page 17, bottom). Abrogation of such Mst1-mediated C/EBP- β phosphorylation in FoxO1-KD cells or in the presence of an NLS mutant FoxO1 will be required to confirm this conclusion.

We conducted additional experiments and the data obtained with FoxO1-KD cells has been added in **Figure 5G**. We confirmed that Mst1-mediated C/EBP- β phosphorylation was abolished in FoxO1-KD cells or in the presence of a FoxO1-NLS mutant.

In addition, using a WT or phosphor-mimic C/EBP- β , the authors stated that Mst1-mediated phosphorylation of FoxO1 and C/EBP- β promotes formation of a trimeric (FoxO1-C/EBP- β -Mst1) complex (Fig. 5B) and C/EBP- β homodimer (Fig. 5F). To support these bold conclusions, experiments using an Mst1-DN and C/EBP- β -PR construct will be necessary in panel B and F, respectively. Homodimer formation of C/EBP- β -PM should be tested in the absence of Mst1.

We conducted additional experiments and the data obtained with Ad-DN-Mst1 and Ad-C/EBP- β -PR has been added in **Figure 5B**, while the data obtained with Ad-C/EBP- β -PR and C/EBP- β -PM in the absence of Mst1 has been added in **Figure 5F**.

5. In Figure 7, using a novel C/EBP- β -KI mouse model, the authors have demonstrated the importance of C/EBP- β phosphorylation to attenuate ischemia/reperfusion-induced myocardial damage (Fig. 7A). In Figures 6 and Fig. S12, the authors have also utilized different transgenic and knockout mouse models and demonstrated that loss of Mst1 or FoxO1 activity trigger severe myocardial damage following prolonged ischemia (Fig. 6E) and injection of phosphor-mimic C/EBP- β virus into the LV tissue of FoxO1-cKO mice attenuated ischemic damage (Fig. S12A-B). To strengthen the novelty of these findings reciprocal experiments using C/EBP- β -cKO mice (Fig. S7B) and injection of FoxO1 virus will be essential. In addition, unlike FoxO1-PR, it is not clear why overexpression of FoxO1-WT in cardiomyocytes in vitro (Fig. 3A) and in vivo (Fig. S12) had dramatic inverse effects on cell death.

The reviewer raised important issues.

As suggested by the reviewer, we conducted the reciprocal experiments. We tested the effect of exogenous FoxO1 upon cardioprotection in mice with a loss of C/EBP- β function. The new results show that the protective effect of FoxO1 *in vivo* is abolished in C/EBP- β cKO mice, suggesting that endogenous C/EBP- β plays an important role in mediating the

protective effect of FoxO1, consistent with the existing results and our hypothesis (New **Figure S11A**).

Regarding the inverse effects of FoxO1-WT *in vitro* and *in vivo*, it may be that the level of Mst1 activity affects whether the overall effect of FoxO1 becomes either pro- or anti-apoptotic. *In vitro*, where there is less Mst1, the anti-apoptotic effect of endogenous FoxO1 through C/EBP- β would be weaker and, thus, the pro-apoptotic effect of FoxO1 would become more prominent. *In vivo*, where Mst1 is active at baseline, the pro-apoptotic effect of endogenous FoxO1 may be negatively affected by the anti-apoptotic effect of FoxO1 through C/EBP β .

Minor Points:

1. *The figure legend and data are different for Figure 4B.*

We corrected the figure legend for **Figure 4B**.

2. *Data for FASLG and PMAIP1 are missing in Fig. S5B.*

We added the immunoblot data for FASLG and PMAIP1 in new **Figure S8A**.

Reviewer #3 (Remarks to the Author):

Major comments

1. *In the results section- titled Mst1 physically interacts with and phosphorylates FoxO1, thereby enhancing nuclear translocation of FoxO1 in CMs, paragraph ending with “Mst1-overexpressing CMs, FoxO1 failed to interact with 14-3-3 proteins”. The authors focus on the role of MST1 in regulating FoxO1 binding to 14-3-3. Du X et al., (J Immunol 2014) have already demonstrated that MST1 stabilizes both FoxO1 and 3 by inhibiting Akt activity. Please comment on the endogenous basal Akt levels and phospho-Akt levels under MST1 activation? Is the inability of 14-3-3 to interact with FoxO1 an Akt mediated effect?*

As suggested by the reviewer, we investigated the effect of Mst1 on endogenous Akt activity through immunoblotting. We found that Mst1 overexpression suppressed Akt phosphorylation. Conversely, Akt phosphorylation was increased when Mst1 activity was inhibited by overexpressing DN-Mst1 (New **Figure S1E**). Taken together, the inability of 14-3-3 to interact with FoxO1 may be in part dependent on the effect of Mst1 upon changes in Akt activity (**Figure S1D**). Thank you for your suggestion.

2. *The results section entitled - Mst1 protects FoxO1 from degradation but inhibits its DNA binding in CMs, ending in page 11 with sentence – “The results suggest that although Mst1-mediated FoxO1 phosphorylation protects FoxO1 from protein degradation, it attenuates the DNA binding ability of FoxO1, thereby suppressing the transcriptional activity of FoxO1 on the DBE”. The focus again has been completely on MST1. It could be possible that MST1 interaction with FoxO1 DBD could be responsible for preventing transcriptional activity but the prevention of FoxO1 ubiquitination and degradation could be via MST1 mediated inhibition of*

Akt. The endogenous Akt 1 total and phosphorylation status along with the activity should be measured.

As described above, the reviewer is correct in that Mst1-induced suppression of Akt alone may explain nuclear translocation of FoxO1 through inhibition of 14-3-3 binding to FoxO1 (New **Figure S1D**). However, as we described in our response to Reviewer #2 (Major Question 1), in the newly conducted experiments in this revision, we discovered that the molecular mechanism mediating the effect of Mst1 upon FoxO1 translocation and that mediating the effect of Mst1 upon dissociation of FoxO1 from the conventional FoxO1 DNA binding sequence are not identical. Thus, we speculate that, although Mst1-induced nuclear translocation of FoxO1 may be explained in part by the effect of Mst1 upon Akt (phosphorylation and inhibition of Akt), the other effect of Mst1 (inducing dissociation of FoxO1 from the conventional FoxO1 binding site) may not.

3. In the results section titled – “Co-expressed FoxO1 and Mst1 significantly reduced H2O2-induced apoptosis in CMs by upregulating antioxidant genes and downregulating pro-apoptotic genes”, the authors claim that MST1 regulation of FoxO1 is responsible for upregulating anti-oxidant genes and downregulating pro-apoptotic genes. If MST1 binding prevents the DBD-FoxO1 to bind to its responsive elements, how can they turn on the anti-oxidant genes?

Based on the findings of our current study, phosphorylation of FoxO1 by Mst1 abolishes its function as a transcription factor and instead confers an alternative role as an adaptor for C/EBP- β to be phosphorylated by Mst1, dimerized and activated. ChIP-sequencing analyses showed that C/EBP- β phosphorylated by Mst1 competes with endogenous FoxO1 for binding to the FoxO1 sites. *We speculate that displacement of FoxO1 with phosphorylated C/EBP- β may take place in some antioxidant genes in which the C/EBP- β DNA binding site is near the FoxO1 binding site.* We discovered several genes encoding antioxidant proteins that share a similar characteristic with the catalase gene (**Table S2**).

4. If the anti-oxidant genes are a result of CEBP - beta activation, then the MST1-FoxO1 nexus becomes redundant rather finding the direct link between activation of CEBP-beta by MST1 would be a better strategy. In the FoxO1 knock down cells, and in cardiac FoxO1 knock out animals, test whether MST1 activation could lead to CEBP - beta activation and enhancement of transcription function with resultant upregulation of antioxidant genes.

This is a wonderful suggestion. We have now investigated how phosphorylation of C/EBP- β is affected in the absence of FoxO1, even when Mst1 is activated (New **Figure 5H**). We found that Mst1-induced phosphorylation of C/EBP- β at Thr299 does not take place when FoxO1 is downregulated. As described in #3 above, the new result further strengthens our hypothesis that *FoxO1 acts as an adaptor and plays an essential role in mediating Mst1-induced phosphorylation of C/EBP- β and its consequent activation.*

5. There is a high level of divergence in the models used. The authors have moved from using H2O2 to increase oxidative stress and activate MST1 and in latter figures have employed hypoxia model and ischemic model (in vivo) for confirming CEBP- beta’s role in protection. The continuity is lost in translation and it is important to provide evidence whether MST1 is the master regulator of CEBP-beta in all of the models used. The authors also need to explain whether this CEBP-beta regulation is FoxO1 dependent or independent?

We fully understand the reviewer's concerns. We wish to respectfully point out that, although we investigated the role of Mst1 in multiple models in this study, we have shown previously that all stimuli used in the current investigation induce myocardial oxidative stress and activate Mst1 (Yamamoto S. JCI 2003;111:1463, Odashima M. Circ Res 2007;100:1344, Maejima Y. Nat Med 2013;19:1478, Del Re DP. Mol Cell 2014;54:639, Sciarretta S. Cell Rep 2015;11:125, Matsuda T. Circ Res 2016;119:596, etc.). Many investigators in the field use oxidative stress as a surrogate for myocardial ischemia and ischemia/reperfusion. We routinely show the *in vivo* relevance of *in vitro* observations with mouse models of ischemia and ischemia/reperfusion.

6. *It could also be possible that other FoxO family members playing a role in anti-oxidant genes upregulation as shown in Figure-4. Measure the total and nuclear fractions and DNA-binding abilities of FoxO3, and FoxO4 in the presence of MST1 and DN-MST1 conditions.*

We assessed the effect of Mst1 on the total amount and nuclear distribution of FoxO3 and FoxO4 by conducting immunoblotting. The total protein levels of both FoxO3 and FoxO4 remained unchanged by either the presence or absence of Mst1 activation. Mst1 facilitated the nuclear translocation of both FoxO3 and FoxO4. Conversely, suppression of Mst1 activity with DN-Mst1 inhibited the nuclear translocation of both FoxO3 and FoxO4 (**Figure S1BC**). Since the sequence of the DNA binding domain is conserved among FoxO1, FoxO3, and FoxO4, the impact of Mst1 activity on the DNA-binding abilities of these genes may be identical. FoxO1, FoxO3 and FoxO4 possess some, though not all, redundant functions in the heart. We now discuss this issue as an experimental limitation (Page 30, line 3 from the bottom - Page 31, line 3).

7. *Under conditions of hypoxia or ischemia (both in vitro and in vivo) conditions, a major transcription factor that could be playing a parallel role is HIF1 alpha. Under the given conditions, it is important to measure HIF1 alpha expression in total and nuclear compartments? Also, it is important to measure whether CEBP-beta is complexing in the nucleus with HIF1 alpha along with FoxO1 and other FoxO members.*

We agree with the reviewer. We and others have shown that HIF-1 α plays a protective role in the heart during ischemia and ischemia/reperfusion. Although it would be interesting to extend our study to investigate the involvement of HIF-1 α in mediating the protective mechanism of Mst1, this would require many new experiments. Thus, we would like to pursue this avenue as a new project in the near future.

Minor comments:

1. *In the introduction part, second page first paragraph – states stimulate physical interaction between Beclin1 and the Bcl-2 family proteins, thereby inhibiting autophagy. Which Bcl2 family member is being alluded to?*

In our previous paper (Maejima Y. Nat Med 2013), we discovered that Beclin1, phosphorylated by Mst1, physically interacts with Bcl-2 and Bcl-xL.

2. *In the introduction part- second page second paragraph states that - on the other hand, several other kinases can promote FoxO activation by facilitating the nuclear translocation of FoxO. The authors have loosely mentioned several kinases- need to specify which kinases are being discussed. Also, when the authors mention FoxO activation and translocation- which FoxO family member are they referring to?*

AMPK induces nuclear translocation and activation of FoxO1 and FoxO3 (Page 4, lines 12-13; Ref. 12). JNK induces nuclear translocation and activation of FoxO4 (Page 4, lines 13-16; Ref. 13).

3. *Neonatal cardiomyocytes being used in this study may perform a few divisions as compared to adult myocytes that are terminally differentiated and do not undergo cell-division. Please comment.*

The reviewer is correct. We respectfully point out that our major *in vitro* findings were confirmed in adult mice *in vivo*. We include a discussion of this issue as an experimental limitation of the study (Page 31, lines 3-5).

REVIEWERS' COMMENTS

Reviewer #1 (Remarks to the Author):

Thanks for addressing my concerns by generating and including new data.

Reviewer #2 (Remarks to the Author):

The authors have addressed my concerns.

Reviewer #3 (Remarks to the Author):

No additional comments.

Point by point responses to the reviewers

We thank the reviewers for favorable comments.

Reviewer #1:

Thanks for addressing my concerns by generating and including new data.

Our response: Thank you very much.

Reviewer #2:

The authors have addressed my concerns.

Our response: Thank you very much.

Reviewer #3:

No additional comments.

Our response: Thank you very much.